# FR: Folded Rationalization with a Unified Encoder

**Wei Liu**[1]    **Haozhao Wang**[1*]  **Jun Wang**[2*]  **Ruixuan Li**[1*]  **Chao Yue**[1]   **Yuankai Zhang**[1]

[1]School of Computer Science and Technology, Huazhong University of Science and Technology
[2]iWudao Tech

[1]`{idc_lw, hz_wang, rxli, yuechao, yuankai_zhang}@hust.edu.cn`
[2]`jwang@iwudao.tech`

## Abstract

Conventional works generally employ a two-phase model in which a generator selects the most important pieces, followed by a predictor that makes predictions based on the selected pieces. However, such a two-phase model may incur the degeneration problem where the predictor overfits to the noise generated by a not yet well-trained generator and in turn, leads the generator to converge to a sub-optimal model that tends to select senseless pieces. To tackle this challenge, we propose Folded Rationalization (FR) that folds the two phases of the rationale model into one from the perspective of text semantic extraction. The key idea of FR is to employ a unified encoder between the generator and predictor, based on which FR can facilitate a better predictor by access to valuable information blocked by the generator in the traditional two-phase model and thus bring a better generator. Empirically, we show that FR improves the F1 score by up to $10.3\%$ as compared to state-of-the-art methods. Our codes are available at https://github.com/jugechengzi/FR.

## 1   Introduction

There are growing concerns over the interpretability of NLP models, especially when language models are being rapidly applied on various critical fields (Lipton, 2016; Du et al., 2019; Xiang et al., 2019; Miller, 2019; Sun et al., 2021). Rationalization, using a cooperative game between a generator and a predictor in which the generator selects distinguishable and human-intelligible pieces of the inputting text (i.e., rationale) to the followed predictor that maximizes the predictive accuracy, has become one of the mainstream approaches to improve the interpretability of NLP models. The standard rationalization method named RNP (Lei et al., 2016) organizes the generator and predictor with a two-phase framework (see Figure 2(a)). However, as illustrated in Table 1, such a two-phase model suffers from the degeneration problem where the predictor may overfit to meaningless but distinguishable rationales generated by the not yet well-trained generator (Yu et al., 2019), leading the generator to converge to the sub-optimal model that tends to select these uninformative rationales.

Many approaches have been proposed to address the degeneration issue. The basic idea of these approaches is to regularize the predictor using supplementary modules that make use of the full text such that the predictor does not rely entirely on the rationale provided by the generator. For example, as shown in the Figure 2, 3PLAYER (Yu et al., 2019) adopts an extra predictor to squeeze informative parts from the unselected text pieces into the rationale; DMR (Huang et al., 2021) additionally aligns with prediction distribution and feature distribution of of the full text; A2R (Yu et al., 2021) combines binary selection with soft selection in which every token in the inputting text is partly contained.

---

*Corresponding authors. This paper is a collaboration between Intelligent and Distributed Computing Laboratory, Huazhong University of Science and Technology and iWudao Tech.

Table 1: An example of RNP making the right prediction using the uninformative rationale. The underlined piece of the text is the human-annotated rationale. Pieces of the text in red and blue represent the rationales from RNP and our method respectively. Initially, the generator may randomly select some uninformative rationales such as selecting "-" for the negative text. The predictor of RNP overfits to these uninformative rationales and recognizes the category according to whether "-" is included in the rationale. Guided by such a spoiled predictor, the generator in turn tends to select these uninformative rationales.

| |
|---|
| **Label(Aroma):** Negative |
| **Input text:** 12 oz bottle poured into a pint glass - a - pours a transparent , pale golden color . the head is pale white with no cream , one finger 's height , and abysmal retention . i looked away for a few seconds and the head was gone s - stale cereal grains dominate . hardly any other notes to speak of . very mild in strength t - sharp corn/grainy notes throughout it 's entirety . watery , and has no hops characters or esters to be found. very simple ( not surprisingly ) m - highly carbonated and crisp on the front with a smooth finish d - yes , it is drinkable , but there are certainly better choices , even in the cheap american adjunct beer category . hell , at least drink original coors |
| **Rationale from RNP:** ["-"]  **Pred:** Negative |
| **Rationale from FR:** ["stale cereal grains dominate . hardly any other notes to"]  **Pred:** Negative |

Although these regularized methods can greatly alleviate the degeneration problem, they still maintain the two fragmented phases in which the spoiled predictor can only be partially calibrated by the extra modules. More specifically, we identify that the predictor has notably higher learning speed than the generator under the two-phase framework, indicating that a regularized predictor will not immediately lead to a well-trained generator and has to continuously learn from generated uninformative rationales until convergence. To show this, we consider the RNP method with different learning rates for the generator and predictor. We defer the experimental details to Appendix B.2 and present the results of rationale quality (F1 score) in Figure 1. As can be seen, the results usually get better when the learning rate of the predictor is smaller than the generator. In particular, the result achieves the best when the learning rate of the predictor is approximately $1/5$ that of the generator, where the learning speed of the predictor is exceptionally limited.

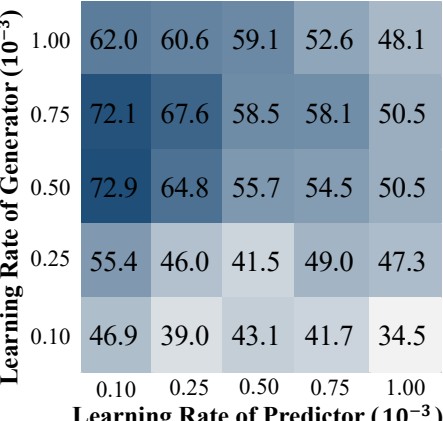

Figure 1: The rationale quality of RNP with different learning rates for the generator and the predictor.

Although balancing the learning speed between the generator and predictor can effectively improve the performance, tuning the learning rates involves careful coordination between the two phases and requires much human effort, especially when some existing methods are designed with more modules. To tackle this challenge, we in this work propose a frustratingly simple but effective method, Folded Rationalization (FR), which folds the two phases of the current rationalization method into one using a unified encoding mechanism. Specifically, FR shares a unified text encoder between the generator and the predictor. The predictor is enforced with the same learning speed as the generator and has significant chances of learning informative rationales from a well-trained generator. On the other hand, the generator can also get a higher learning speed with the auxiliary of the predictor. Additionally, as theoretically analyzed, the encoder of the generator can also be seen as a special regularizer for the predictor, which helps it get a global view of all the possible rationale candidates from the full text and thus prohibits it from overfitting to the uninformative ones. Furthermore, we evaluate our approach on two widely used rationalization benchmarks, i.e., the Beer Reviews dataset (McAuley et al., 2012) and the Hotel Reviews dataset (Wang et al., 2010), of which the empirical results show that FR outperforms all state-of-the-art methods in terms of the rationale quality. Our contributions are:

• To the best of our knowledge, this paper is the first to solve the degeneration problem in rationalization from the perspective of re-organizing the two-phase framework. We propose a simple but

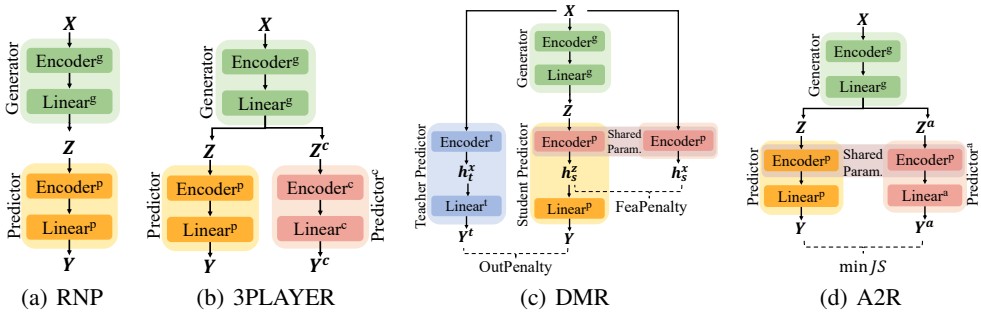

Figure 2: Standard method (a) RNP (Lei et al., 2016) with other recent methods with additional regularized modules. (b) 3PLAYER (Yu et al., 2019). (c) DMR (Huang et al., 2021). (d) A2R (Yu et al., 2021). $X, Z, Y$ represent the input text, selected rationale, and predictive result, respectively.

effective method, namely, FR, that folds the two phases into one by sharing a unified encoder between the generator and the predictor.

• The degeneration problem is mainly because the predictor can make correct predictions using uninformative rationales, leading to a sub-optimal generator. In this paper, we theoretically prove that the predictor of FR will not make correct predictions using these uninformative ones and thus mitigate the degeneration problem.

• We conduct extensive experiments over various datasets. The performance superiority of FR is demonstrated as compared to traditional rationalization methods. In particular, the F1 score of FR can be up to $10.3\%$ higher than the state-of-the-art method.

## 2 Related work

**Optimization for neural rationales.** Most current rationalization methods adopt a cooperative framework between generator and predictor (Lei et al., 2016), which requires careful optimization to coordinate and is hard to train. Considering this challenge, a series of research efforts focus on refining the optimization process to improve the rationalization. For instance, Bao et al. (2018) used Gumbel softmax to do the reparameterization. Bastings et al. (2019) replaced the Bernoulli sampling distributions with rectified Kumaraswamy distributions. Chang et al. (2019) introduced a adversarial game and produces both positive and negative rationales. Jain et al. (2020) disconnected the training regimes of the generator and predictor networks using saliency threshold. Chang et al. (2020) tried to learn the invariance in rationalization. Jiang et al. (2021) used a post-hoc local approach to generate rationales for well-trained models. These methods are orthogonal to our proposed method FR that shares an encoder between the generator and predictor.

**Regularization for neural rationales.** Another series of efforts seek to regularize the predictor using supplementary modules which have access to the information of the full text (Yu et al., 2019; Sha et al., 2021; Huang et al., 2021; Yu et al., 2021). Figure 2 shows a comparison of some representative methods, where $X, Z, Y$ represent the input text, the generated rationale, and the predictive output, respectively. Yu et al. (2019) took the unselected text $Z^c$ into consideration by employing a supplementary predictor *Predictor$^c$*, as shown in Figure 2(b). The generator and the supplementary predictor play an adversarial game. *Predictor$^c$* tries to maximize the predictive accuracy while the *Generator* seeks to minimize the predictive accuracy of *Predictor$^c$* such that all the informative pieces is squeezed into $Z$ from $Z^c$. Huang et al. (2021) tried to align the distributions of rationale with the full input text in both the output space and feature space, as shown in Figure 2(c). To make alignment in the output space, they introduce a teacher predictor *Predictor$^t$* pre-trained with the full text and minimize the cross-entropy between the distributions of $Y$ and $Y^t$ (*OutPenalty*). In feature space, they minimize the central moment discrepancy (*FeaPenalty*) (Zellinger et al., 2017) between the representation of $X$ and $Z$. Yu et al. (2021) endowed the predictor with the information of full text by introducing a soft rationale, which is shown in Figure 2(d). Specifically, the supplementary predictor takes the masked text by soft attention $Z_{\text{soft}}$ as the input. To convey the formation of full text to the predictor, the model minimizes the JS-divergence between the output $Y$ of the original predictor and $Y^s$ of the supplementary predictor. These methods are mostly related to our work.

However, different from these methods, we in this paper solve the degeneration problem from a new perspective. We fold the two phases of the rationalization framework into one such that the predictor gets direct access to the original full text.

## 3 Problem definition

**Notation** In the following sections, we use $\mathbb{P}(\cdot)$ and $p(\cdot)$ to represent distribution and probability, respectively. $gen(\cdot)$ and $pred(\cdot)$ represent the generator and predictor, respectively. $\theta_g$ and $\theta_p$ represent the parameters of the generator and predictor, respectively. $\mathcal{D}$ represents the distribution of dataset. We consider the classification problem, where the input is a text sequence $X=[x_1, x_2, \cdots, x_l]$ with $x_i$ being the $i$-th token and $l$ being the number of tokens. The label of $X$ is a one-hot vector $Y \in \{0, 1\}^c$, where $c$ is the number of categories.

**Cooperative rationalization** Cooperative rationalization framework consists of a generator and a predictor. The goal of the generator is to find the most informative pieces containing several tokens in the original input text $X$. For each sample $(X, Y) \sim \mathcal{D}$, the generator firstly outputs a sequence of binary mask $M = [m_1, \cdots, m_l] \in \{0, 1\}^l$. Then, it forms the rationale $Z$ by the element-wise product of $X$ and $M$:

$$Z = M \odot X = [m_1 x_1, \cdots, m_l x_l]. \tag{1}$$

In cooperative rationalization, the informativeness of the rationale $Z$ provided by the generator is measured by the negative cross entropy $-H(Y, Y_z)$, where $Y_z$ is the output of the predictor with the input being $Z$. Consequently, the generator and the predictor are usually optimized cooperatively:

$$\min_{\theta_g, \theta_p} \sum_{(X,Y)\sim\mathcal{D}} H(Y, pred(gen(X))). \tag{2}$$

**Regularizer of shortness and coherence** Instead of using more complicated regularizers in the other models trying to mitigate degeneration (Yu et al., 2019; Huang et al., 2021), our model only needs a basic regularizer (Chang et al., 2019) as follows, which is similar to RNP (Lei et al., 2016), to effectively control the sparsity and coherence of the selected rationales:

$$\Omega(M) = \lambda_1 |\frac{||M||}{l} - \alpha| + \lambda_2 \sum_t |m_t - m_{t-1}|, \tag{3}$$

where $l$ denotes the number of tokens in the input text. The first term encourages that the percentage of the tokens being selected as rationales is close to a pre-defined level $\alpha$. The second term encourages the rationales to be coherent.

## 4 Folded rationalization using unified encoder

In this section, we specify FR and analyze why it works both intuitively and formally.

### 4.1 Architecture of folded rationalization

Based on the framework of RNP, the architecture of FR is to share the encoder between the generator and the predictor, as shown in Figure 3. It is worthwhile to note that we have no constraints on the types of the encoder and it can be any models such as RNN and Transformer (Vaswani et al., 2017). For the convenience of comparing with current methods in experiments, in this paper, we adopt the bidirectional gated recurrent units (GRU) (Cho et al., 2014) as the encoder which has been adopted by most previous works (Huang et al., 2021; Yu et al., 2021). In FR, the full text $X$ is first sent to the unified *Encoder* in the generator and it outputs a representation for each token in $X$. Then, the representation of each token is independently sent to *Linear$^g$* which is a linear layer acting as the generator head. *Linear$^g$* outputs a value of probability following the Bernoulli distribution from which the mask of each token is independently sampled using gumbel-softmax (Jang et al., 2016). After that, the generator outputs $Z$ using equation 1 and further sends it to the unified encoder *Encoder* in the predictor. By pre-processing the output of *Encoder* with max-pooling, the predictor uses the linear layer *Linear$^p$* to make the prediction. We adopt the similar objective function as RNP, i.e., equation 2 and 3, to optimize the generator and predictor cooperatively.

**Intuition for guaranteed predictive accuracy** By taking the classification problem as example, we here discuss why FR will not reduce the predictive accuracy of the predictor. In principle, the generator and the predictor seek to extract the same semantic information from the similar input text. Specifically, to select the most informative pieces from the original text, the generator has to extract the semantic information of the text which is usually specific and distinguishable from other texts. On the other hand, to make the correct prediction, the predictor needs to transform the selected pieces mostly relative to the input text into the semantic features which are distinguishable and separable. Therefore, a unified

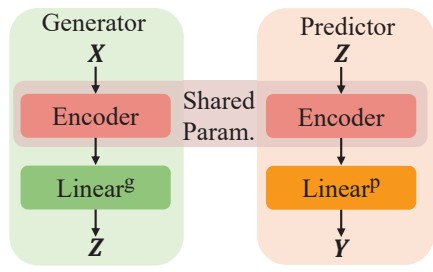

Figure 3: The architecture of FR. $X$ is the original full text. $Z$ is the rationale and $Y$ is the predictor's output.

encoder will not degrade the performance of the predictor, but will encourage them to benefit from each other.

**Mutually reinforcing generator and predictor** The main problem in RNP is that both the generator and the predictor only have access to partial information of the original data pairs, which requires careful coordination to obtain the missed information from each other. FR with a unified encoder provides inherent benefits for the coordination and promotes the information sharing. To be more specific, the generator has access to the input text but no true labels, while the predictor is on the contrary. To achieve good results, they have to carefully make a coordination such that receive valuable information from one another. However, there are $2^l$ candidate rationales for each $X$ with length $l$, which is exceptionally difficult for a not well trained generator to select informative ones from such a large space. As a consequence, the predictor of RNP has high risks in overfitting the uninformative noise provided by the generator. There is a similar case for the generator as obtaining backward information from the predictor. FR addresses this issue by allowing the generator and predictor to directly access to the full information of the original data. Specifically, the predictor with the unified encoder has a global view of all the rationale candidates by direct access to $X$, which prevents itself from overfitting to uninformative ones. In turn, with the encoder of the predictor, the generator's ability of summarizing and extracting information from the original text is greatly enhanced.

### 4.2 Theoretical analysis

In the cooperative rationalization, the generator uses the predictor to evaluate the informativeness of rationales. If the predictor can always classify uninformative rationales into true labels, then the generator may be fooled and converge to be sub-optimal. Here we will show that the predictor in FR is regularized by the generator through the unified encoder and can not stably classify uninformative rationales into true labels any more. In this way, only informative rationales can promise high predictive accuracy. To this end, once the equation 2 is optimized to maximize the final predictive accuracy, the selected rationales is guaranteed to be informative ones. To quantify the uninformativeness, we first give a definition of uninformative tokens:

**Definition 1** *For $X \sim \mathcal{D}$, $X_s$ is a subset of $X$, for a given token $t$, we say it is uninformative for $X$ if the following equation holds:*

$$\mathbb{P}(Y|X_s) = \mathbb{P}(Y|X_s, t), \quad s.t., \ \forall X_s \subset X \tag{4}$$

It means that token $t$ is independent of $Y$ given $X_s$. This definition comes from how the Shapley value is calculated for no contribution, i.e., all marginal contributions are zero. In this situation, we can say $t$ is uninformative to $Y$ for $X$ naturally. When equation 2 gets the optimal solution, we have $pred(gen(X)) = p(Y|X)$ and $pred(gen([X, t])) = p(Y|X, t)$, where "$[X, t]$" denotes the result of adding token $t$ to any position of $X$. Then, we derive the following lemma.

**Lemma 1** *When an uninformative token $t$ is added to the original text $X$, the predictive results will hardly change:*

$$pred(gen(X)) = pred(gen([X, t])) + \epsilon, \ s.t., \ \forall X_s \subset X, \ \mathbb{P}(Y|X_s) = \mathbb{P}(Y|X_s, t), \tag{5}$$

*where $\epsilon$ is an error depending on the learning error of the neural network. When equation 2 gets the optimal solution, the absolute value of $\epsilon$ is $0$.*

Without losing generality, we consider the case when equation 2 achieves the optimal solution for simplicity. For ease of exposition, we here denote $gen(X)$ and $gen([X, t])$ as $Z(X)$ and $[Z_1(X), Z_2(t)]$ respectively, where $Z_1(X)$ represents the binary masks of tokens in $X$ and $Z_2(t)$ represents the mask of the new token $t$. Using equation 5 and $\epsilon = 0$, we have $pred(Z(X)) = pred([Z_1(X), Z_2(t)])$. To promise the same prediction, we get $Z(X) = Z_1(X)$, which indicates

$$gen(X) = \overline{gen([X, t])}, \; s.t., \; \forall X_s \subset X, \; \mathbb{P}(Y|X_s) = \mathbb{P}(Y|X_s, t), \tag{6}$$

where $\overline{gen([X, t])}$ denotes the first item of generator's output, e.g., $Z_1(X)$. For convenience of analyzing the functionality of the unified encoder, we decompose the generator and the predictor as

$$gen(\cdot) = f_g \circ h_g, \; pred(\cdot) = f_p \circ h_p, \tag{7}$$

where "$\circ$" stands for functional composition, i.e., $(f \circ h)(X) = f(h(X))$. $h$ and $f$ represent the encoder and linear layer, respectively. Obviously, we have $h_g = h_p$ for FR. We rewrite equation 6 as $f_g \circ h_g(X) = \overline{f_g \circ h_g([X, t])}$. Since $f_g$ is a linear function, and the mask is selected independently for each token in practice, we get $h_g(X) = \overline{h_g([X, t])}$. Now, we have the following lemma.

**Lemma 2** *If equation 2 gets the optimal solution, adding an uninformative token $t$ to the text $X$ does not change the representation of the original tokens in $X$ output by the encoder of the generator, i.e.,*

$$h_g(X)_{x_i} = h_g([X, t])_{x_i}, \; \forall x_i \in X, \tag{8}$$

*where $h_g(X)_{x_i}$ represents the token $x_i$'s representation with $X$ being the input of the encoder $h_g$.*

In the following, we consider $h_g$ as a general one-way RNN network. Note that bidirectional RNN is almost the same. Besides, our analysis also holds for Transformer encoder which we leave its proof in the Appendix A.2. For the $i$-th token in $X$, the representation can be expressed as:

$$h_g(X)_{x_i} = \varphi_g(s_i), \; s_i = \psi_g(e_i, s_{i-1}), \tag{9}$$

where $e_i, s_i$ are the word embedding and hidden state of the $i$-th token. $\varphi_g, \psi_g$ are two functions corresponding to the type of RNN units. When an uninformative token $t$ is added between $x_i$ and $x_{i-1}$, we have

$$s_t = \psi_g(e_t, s_{i-1}), \; s_i' = \psi_g(e_i, s_t), \tag{10}$$

where $s_i'$ is the hidden state of the original $x_i$ in this new text and we denote it by $h_g([X, t])_{x_i'} = \varphi_g(s_i')$. From Lemma 2, we have $h_g(X)_{x_i} = h_g([X, t])_{x_i'}$. Using the first term of of the first equation 9, we get

$$\varphi_g(s_i) = h_g(X)_{x_i} = h_g([X, t])_{x_i'} = \varphi_g(s_i'). \tag{11}$$

Then we easily get $s_i = s_i'$. Using the second term of equation 10, we have

$$\psi_g(e_i, s_t) = s_i' = s_i = \psi_g(e_i, s_{i-1}). \tag{12}$$

Then we have $s_t = s_{i-1}$ very smoothly, which is specified in the following lemma.

**Lemma 3** *If the $i$-th token in $X$ is uninformative, the hidden state of it through the encoder of generator is equal to that of the token preceding it:*

$$s_i = s_{i-1}, \; s.t., \; \forall X_s \subset X, \; \mathbb{P}(Y|X_s) = \mathbb{P}(Y|X_s, x_i). \tag{13}$$

Then, let us consider two different uninformative rationales, i.e., $Z_{uninf}$ and $Z_{uninf}'$, which contain only uninformative tokens and come from two different data pairs $(X, Y), (X', Y')$.

**Lemma 4** *If equation 13 holds, any two different uninformative rationales have the same representation through the encoder of generator for each token, i.e., $h_g(Z_{uninf})_{z_i} = h_g(Z_{uninf}')_{z_j}$, where $h_g(Z_{uninf})_{z_i}$ denotes the representation for the $i$-th token in $Z_{uninf}$.*

The proof can be seen in Appendix A.1. Now, we immediately get the following theorem.

**Theorem 1** *If FR gets the optimal solution of equation 2, i.e., $\mathbb{P}(Y|X) = \mathbb{P}(pred(gen(X)))$, and $Z_{uninf}, Z_{uninf}'$ are two different rationales that contain only uninformative tokens and are selected from two different data pairs $(X, Y), (X', Y')$, we have*

$$(f_p \circ h_g)(Z_{uninf}) = (f_p \circ h_g)(Z_{uninf}'). \tag{14}$$

Theorem 1 indicates that the predictor in FR will give the same output for any uninformative rationales. Hence, when we optimize the generator and the predictor by maximizing the final predictive accuracy using equation 2, the predictor only recognizes the informative rationales and ignores the uninformative ones. In this way, our FR greatly mitigates the degeneration problem.

Table 2: Results on Beer Reviews and Hotel Reviews. Each aspect is trained independently.

(a) Beer Reviews

| Methods | Appearance | | | | | Aroma | | | | | Palate | | | | |
|---|---|---|---|---|---|---|---|---|---|---|---|---|---|---|---|
| | S | Acc | P | R | F1 | S | Acc | P | R | F1 | S | Acc | P | R | F1 |
| RNP | 18.7 | 84.0 | 72.0 | 72.7 | 72.3 | 15.1 | 85.2 | 59.0 | 57.2 | 58.1 | 13.4 | 90.0 | 63.1 | **68.2** | 65.5 |
| DMR | 18.2 | - | 71.1 | 70.2 | 70.7 | 15.4 | - | 59.8 | 58.9 | 59.3 | 11.9 | - | 53.2 | 50.9 | 52.0 |
| A2R | 18.4 | 83.9 | 72.7 | 72.3 | 72.5 | 15.4 | 86.3 | 63.6 | 62.9 | 63.2 | 12.4 | 81.2 | 57.4 | 57.3 | 57.4 |
| FR(ours) | 18.4 | 87.2 | **82.9** | **82.6** | **82.8** | 15.0 | 88.6 | **74.7** | **72.1** | **73.4** | 12.1 | 89.7 | **67.8** | 66.2 | **67.0** |

(b) Hotel Reviews

| Methods | Location | | | | | Service | | | | | Cleanliness | | | | |
|---|---|---|---|---|---|---|---|---|---|---|---|---|---|---|---|
| | S | Acc | P | R | F1 | S | Acc | P | R | F1 | S | Acc | P | R | F1 |
| RNP | 8.8 | 97.5 | 46.2 | 48.2 | 47.1 | 11.0 | 97.5 | 34.2 | 32.9 | 33.5 | 10.5 | 96.0 | 29.1 | 34.6 | 31.6 |
| DMR | 10.7 | - | 47.5 | **60.1** | 53.1 | 11.6 | - | 43.0 | 43.6 | 43.3 | 10.3 | - | 31.4 | 36.4 | 33.7 |
| A2R | 8.5 | 87.5 | 43.1 | 43.2 | 43.1 | 11.4 | 96.5 | 37.3 | 37.2 | 37.2 | 8.9 | 94.5 | 33.2 | 33.3 | 33.3 |
| FR(ours) | 9.0 | 93.5 | **55.5** | 58.9 | **57.1** | 11.5 | 94.5 | **44.8** | **44.7** | **44.8** | 11.0 | 96.0 | **34.9** | **43.4** | **38.7** |

# 5 Experiments

## 5.1 Experimental setup

**Datasets and Models** Following (Huang et al., 2021), we consider two widely used rationalization datasets. To the best of our knowledge, none of dataset contains personally identifiable information or offensive content. The details of the two dataset can be found in Appendix B.1. 1) **Beer Reviews** (McAuley et al., 2012) is a multi-aspect sentiment prediction dataset widely used in rationalization (Lei et al., 2016; Yu et al., 2019; Chang et al., 2019; Huang et al., 2021; Yu et al., 2021). There is a high correlation among the rating scores of different aspects in the same review, making it difficult to directly learn a rationalization model from the original data. Following the previous work (Lei et al., 2016; Huang et al., 2021; Yu et al., 2021), we use the subsets decorrelated by Lei et al. (2016) and binarize the labels as Bao et al. (2018) did. 2) **Hotel Reviews** (Wang et al., 2010) is another multi-aspect sentiment classification dataset. The dataset contains reviews of hotels from three aspects including location, cleanliness, and service. Each review has a rating on a scale of 0-5 stars. We binarize the labels as Bao et al. (2018) did. On both two dataset, we adopt the GRU (Cho et al., 2014) as the encoder which has been adopted by most previous works (Huang et al., 2021; Yu et al., 2021).

**Baselines and implementation details** We compare FR to the cooperative rationalization framework RNP (Lei et al., 2016) and two latest published models that achieve state-of-the-art results: DMR (Huang et al., 2021) and A2R (Yu et al., 2021), both of which have been specified in section 1. For DMR, we adopt its source code and adjust its sparsity constraint to get a sparsity similar to the annotated rationales. For A2R, we re-implement it to do token level selection as other models do. Since DMR and A2R have been validated to outperform 3PLAYER (Huang et al., 2021; Yu et al., 2021), we in this paper ignore its results for clarity of presentation. Following DMR and A2R, we use the 100-dimension Glove (Pennington et al., 2014) as the word embedding and set the hidden dimension of GRU to be 200. We use Adam (Kingma and Ba, 2014) as the optimizer. All the baselines are tuned a lot of times to find the best hyperparameters. All of the models are implemented with PyTorch and trained on a RTX3090 GPU.

**Metrics** All the methods get similar predictive accuracy. Following (Huang et al., 2021; Yu et al., 2021), we mainly focus on the quality of rationales, which is measured by the overlap between the model selected tokens and human-annotated tokens. $P, R, F1$ indicate the precision, recall, and F1 score, respectively. $S$ indicates the average sparsity of the selected rationales, i.e., the percentage of selected tokens to the whole texts. $Acc$ indicates the predictive accuracy on the test set.

## 5.2 Results

**Comparison with baselines on two datasets** Table 2(a) and 2(b) show the results of selected rationales on two datasets. All the methods choose similar percentage of tokens that is close to the human-annotated sparsity by adjusting the sparsity regularization term in equation 3. Obviously, we gain significant improvements in all six aspects. In particular, the F1 score of our method FR is 82.8% in the Beer-Appearance dataset, which improves the state-of-the-art result by up to 10.3%,

Table 3: Examples of generated rationales. Human-annotated rationales are underlined. Rationales from RNP and FR are highlighted in red and blue respectively.

| RNP | FR |
|---|---|
| **Aspect:** Beer-Appearance
**Label:** Negative, **Pred:** Negative
**Text:** appearance : light yellow to almost clear smell : slight hops , but barely smelled like beer taste : little to none , like a rice lager , zero taste mouthfeel : watery and tasteless drinkability : very easy , goes down easier than water . good for drinking games. | **Aspect:** Beer-Appearance
**Label:** Negative, **Pred:** Negative
**Text:** appearance : light yellow to almost clear smell : slight hops , but barely smelled like beer taste : little to none , like a rice lager , zero taste mouthfeel : watery and tasteless drinkability : very easy , goes down easier than water . good for drinking games. |
| **Aspect:** Beer-Aroma
**Label:** Positive, **Pred:** Positive
**Text:** the appearance was nice . dark gold with not much of a head but nice lacing when it started to dissipate . the smell was ever so hoppy with a hint of the grapefruit flavor that 's contained within . the taste was interesting , up front tart grapefruit , not sweet in the least . more like grapefruit rind even . slight hint of hops and seemingly no malt . the mouth feel was crisp , with some biting carbonation . drinkability was easily above average due to the crispness and lack of sweetness . not the usual taste you expect when drinking a fruit beer . in fact this is my favorite fruit beer ever . | **Aspect:** Beer-Aroma
**Label:** Positive, **Pred:** Positive
**Text:** the appearance was nice . dark gold with not much of a head but nice lacing when it started to dissipate . the smell was ever so hoppy with a hint of the grapefruit flavor that 's contained within . the taste was interesting , up front tart grapefruit , not sweet in the least . more like grapefruit rind even . slight hint of hops and seemingly no malt . the mouth feel was crisp , with some biting carbonation . drinkability was easily above average due to the crispness and lack of sweetness . not the usual taste you expect when drinking a fruit beer . in fact this is my favorite fruit beer ever . |
| **Aspect:** Hotel-Cleanliness
**Label:** Negative, **Pred:** Negative
**Text:** we stayed at the holiday inn new orleans french quarter in late april 2012 . the rooms and hotel did not meet my expectations . the property is tired and worn out . my room had trash behind the furniture , rips in the carpet , wallpaper coming off the wall , no toilet lid , dirty tub and sink and a 110 volt out in the bath that was non functional . i noticed a piece of apple in the corner of the elevator the day i checked in and it was still there 3 days later when i checked out . this property is a cash cow and is always rented so they really do n't have to improve it . i should have spent a little extra money to stay at the weston , sheraton or the hilton . live and learn . | **Aspect:** Hotel-Cleanliness
**Label:** Negative, **Pred:** Negative
**Text:** we stayed at the holiday inn new orleans french quarter in late april 2012 . the rooms and hotel did not meet my expectations . the property is tired and worn out . my room had trash behind the furniture , rips in the carpet , wallpaper coming off the wall , no toilet lid , dirty tub and sink and a 110 volt out in the bath that was non functional . i noticed a piece of apple in the corner of the elevator the day i checked in and it was still there 3 days later when i checked out . this property is a cash cow and is always rented so they really do n't have to improve it . i should have spent a little extra money to stay at the weston , sheraton or the hilton . live and learn . |

i.e., 72.5 of A2R. Table 3 shows some visualized examples of rationales from RNP and FR. More rationale examples of DMR and A2R are in Appendix B.4.

**Comparison with baselines in low sparsity** To show the robustness of our FR, we also conduct an experiment where the sparsity of selected rationales is very low, which is the same as CAR (Chang et al., 2019) and DMR. The results are shown in Table 4. The results of RNP and CAR are obtained from (Chang et al., 2019), and the results of DMR are obtained from (Huang et al., 2021). For the aspects of *Aroma* and *Palate*, we still achieve great improvement over other methods. For *Appearance*, our performance is a little bit worse than DMR. The reason may be that when the sparsity is far from the ground truth, many informative tokens are lost and the predictor can not make the right prediction. While DMR uses an adversarial framework same as CAR in practice and the label is used as an additional input to the generator, so it may work better when the sparsity is extremely low.

**Skewed predictor** To show that our FR does not suffer from the degeneration problem, we conduct the same synthetic experiment that deliberately induces degeneration as Yu et al. (2021) did. The details of the experimental setup can be found in Appendix B.3. The results are shown in Table 5. The results of RNP and A2R are obtained from (Yu et al., 2021). For all the settings, we outperform both RNP and A2R. Especially, for *Palate-skew*20, RNP and A2R can not work at all while our FR is still effective. It can be concluded that FR is much more robust than the other two methods in this situation.

Table 4: Results of methods with low sparsity on Beer Reviews. The results of RNP and CAR are obtained from (Chang et al., 2019), and the results of DMR are from (Huang et al., 2021).

| Methods | Appearance | | | | | Aroma | | | | | Plate | | | | |
|---|---|---|---|---|---|---|---|---|---|---|---|---|---|---|---|
| | S | Acc | P | R | F1 | S | Acc | P | R | F1 | S | Acc | P | R | F1 |
| RNP | 11.9 | - | 72.0 | 46.1 | 56.2 | 10.7 | - | 70.5 | 48.3 | 57.3 | 10.0 | - | 53.1 | 42.8 | 47.5 |
| CAR | 11.9 | - | 76.2 | 49.3 | 59.9 | 10.3 | - | 50.3 | 33.3 | 40.1 | 10.2 | - | 56.6 | 46.2 | 50.9 |
| DMR | 11.7 | - | **83.6** | 52.8 | **64.7** | 11.7 | - | 63.1 | 47.6 | 54.3 | 10.7 | - | 55.8 | 48.1 | 51.7 |
| FR(ours) | 12.7 | 83.9 | 77.6 | **53.3** | 63.2 | 10.8 | 87.6 | **82.9** | **57.9** | **68.2** | 10.0 | 84.5 | **69.3** | **55.8** | **61.8** |

Table 5: Results of skewed predictor that induces degeneration on Beer Reviews.

| Aspect | Setting | RNP | | | | A2R | | | | FR(ours) | | | |
|---|---|---|---|---|---|---|---|---|---|---|---|---|---|
| | | Acc | P | R | F1 | Acc | P | R | F1 | Acc | P | R | F1 |
| Aroma | skew10 | 82.6 | 68.5 | 63.7 | 61.5 | 84.5 | **78.3** | 70.6 | 69.2 | 87.1 | 73.9 | **71.7** | **72.8** |
| | skew15 | 80.4 | 54.5 | 51.6 | 49.3 | 81.8 | 58.1 | 53.3 | 51.7 | 86.7 | **71.3** | **68.0** | **69.6** |
| | skew20 | 76.8 | 10.8 | 14.1 | 11.0 | 80.0 | 51.7 | 47.9 | 46.3 | 85.5 | **72.3** | **69.0** | **70.6** |
| Palate | skew10 | 77.3 | 5.6 | 7.4 | 5.5 | 82.8 | 50.3 | 48.0 | 45.5 | 75.8 | **54.6** | **61.2** | **57.7** |
| | skew15 | 77.1 | 1.2 | 2.5 | 1.3 | 80.9 | 30.2 | 29.9 | 27.7 | 81.7 | **51.0** | **58.4** | **54.5** |
| | skew20 | 75.6 | 0.4 | 1.4 | 0.6 | 76.7 | 0.4 | 1.6 | 0.6 | 83.1 | **48.0** | **58.9** | **52.9** |

**Skewed generator**: To further show that our method is not susceptible to degeneration, we design a new synthetic experiment with special initialization that induces degeneration from the perspective of skewed generator. We first pretrain the generator separately using the text classification label as the mask label of the first token. As a result, the predictor will be able to know the category of a rationale merely through whether the first token is selected as part of the rationale or not and may overfit to this positional bias. So before we cooperatively train the generator and predictor together, the generator is initialized with the intentionally misleading pretrained parameters and the predictor is initialized randomly.

We compare the performance of RNP and our FR in the dataset Beer-Palate, because *Palate* is relatively hard to learn as compared to other aspects (Yu et al., 2021). Besides, the performance of RNP and FR is similar in this aspect when the generator is not deliberately skewed. All the hyperparameters we use here are the same as that of the best results in Table 2(a). The results are shown in Table 6. The $k$ in "*skewk*" denotes the threshold of the skew: we pretrain the generator as a special classifier of the first token for a few epochs until its predictive accuracy is higher than $k$. Since the accuracy increases rapidly in the first a few epochs, obtaining a model that precisely achieves the pre-defined accuracy is almost impossible. Therefore, we use "$Pre\_acc$" to denote the actual predictive accuracy of the generator-classifier when the pre-training process stops. Higher "$Pre\_acc$" means easier to degenerate. We find, in this case, both methods get high predictive accuracy while RNP fails to find the human-annotated rationales.

**Failure examples** Table 7 shows some failure cases of our method. Typically, the failures can be summarized as follows. First, the model fails to grasp high-level language phenomena and world background knowledge. For example, in the case of Beer-Palate, the model fails to understand the analogy statement of "almost feels like drinking champagne" and makes the prediction as negative. Second, the model fails to make logical reasoning. The ground truth of hotel cleanliness should be inferred from the word "comfortable". However, our model makes judgments by selecting the rationales such as "breakfast, staff was friendly..., stay", which have little relevance to the cleanliness but are positive. A similar failure also occurs in the case of Beer-Aroma, where the model selects texts with a strong sentiment but uncorrelated with the predefined ground truth.

# 6 Conclusion, limitations, and future work

In this paper, we analyze the relationship between generator and predictor in terms of working mechanism and propose a simple but effective framework to solve the degeneration problem in the two-phase rationalization. The experimental results show that our proposed methods significantly outperform state-of-the-arts in terms of the rationale quality.

One limitation may be that our theoretical analysis mainly focuses on the idealized models which have perfectly fitted the distribution of the dataset. Besides, although sharing the encoder between the generator and predictor is simple, it is limited to the basic RNP which has no extra modules. Existing

Table 6: Results of skewed generator that induces degeneration in the *Palate* aspect of Beer Reviews.

| Setting | RNP | | | | | | FR(ours) | | | | | |
|---------|---------|------|------|------|------|------|---------|------|------|------|------|------|
| | Pre_acc | S | Acc | R | R | F1 | Pre_acc | S | Acc | P | R | F1 |
| skew60.0 | 63.1 | 13.1 | 87.2 | 42.8 | 45.1 | 43.9 | 62.2 | 12.5 | 84.0 | **59.3** | **59.8** | **59.6** |
| skew65.0 | 66.6 | 14.0 | 83.9 | 40.3 | 45.4 | 42.7 | 66.3 | 14.2 | 81.5 | **59.5** | **67.9** | **63.4** |
| skew70.0 | 71.3 | 14.7 | 84.1 | 10.0 | 11.7 | 10.8 | 70.8 | 14.1 | 88.3 | **54.7** | **62.1** | **58.1** |
| skew75.0 | 75.5 | 14.7 | 87.6 | 8.1 | 9.6 | 8.8 | 75.6 | 13.1 | 84.8 | **49.7** | **52.2** | **51.0** |

Table 7: Some failure cases of FR. The underlined piece of the text is the human-annotated rationale. Pieces of the text in blue represent the rationales from FR.

---

**Beer-Palate**
**Label:**positive, **Prediction:**negative
**Input text:** pours a slight tangerine orange and straw yellow . the head is nice and bubbly but fades very quickly with a little lacing . smells like wheat and european hops , a little yeast in there too . there is some fruit in there too , but you have to take a good whiff to get it the taste is of wheat , a bit of malt , and a little fruit flavour in there too almost feels like drinking champagne , medium mouthful otherwise easy to drink , but not somthinf i 'd be trying every night

---

**Hotel-Cleanliness**
**Label:** positive, **Prediction:** positive
**Input text:** my husband and i just spent two nights at the grand hotel francais , and we could not have been happier with our choice . in many ways , the hotel has exceeded our expectation : the price was within our budget , breakfast was included , and the staff was friendly , helpful and fluent in english . as other travelers have mentioned , the hotel is close to the nation metro station , which makes it easy to get around . the room size was just enough to fit two people , but we had a comfortable stay throughout . overall , the hotel lives up to its high trip advisor rating . we would love to stay here again anytime .

---

**Beer-Aroma**
**Label:** positive, **Prediction:** positive
**Input text:** a- amber gold with a solid two maybe even three finger head . looks absolutely delicious , i dare say it is one of the best looking beers i 've had . s- light citrus and hops . not a very strong aroma t-wow , the hops , citrus and pine blow out the taste buds , very tangy in taste , yet perfectly balanced , leaving a crisp dry taste to the palate . m-light and crisp feel with a nice tanginess thrown in the mix . d- could drink this all night , too bad i only have one more of this brew . notes : one of the best balanced and best tasting ipa 's i 've had to date . ipa fans you have to try this one .

---

works have proposed many extra modules to solve the degeneration problem, which demonstrate great effectiveness. Leveraging these extra modules can predictably achieve performance improvement. However, the relationship among multiple modules will become much more sophisticated, simply sharing the generator and predictor may not be the optimal. How to effectively incorporate existing optimized techniques is a challenge for FR.

Besides, the sharing encoder of FR is similar to multi-task learning when training the generator and predictor are viewed as two cooperative tasks. However, existing multi-task learning methods are usually designed for the case where there are multiple output objectives but only one single input sample. How to bridge FR and multi-task learning both intuitively and theoretically, and then improve the rationalization with techniques in multi-task learning, is still unknown. We leave them as future work.

# 7 Acknowledgements

This work is supported by National Natural Science Foundation of China under grants U1836204, U1936108, 62206102, and Science and Technology Support Program of Hubei Province under grant 2022BAA046.

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
