# A  Theorem proofs

## A.1  The proof of Lemma 4 and Theorem 1

Using equation 13, we get that the hidden state of an uninformative token is equal to that of the token precedes it. Considering two specific uninformative rationales $Z = \{$"$mask$", "$t_1$", "$t_2$"$\}$ and $Z^{'} = \{$"$mask$", "$t_3$", "$t_4$"$\}$ where all tokens in them are uninformative, we have:

$$h_g(Z)_{t_1} = h_g(Z)_{t_2} = h_g(Z)_{mask} = h_g(Z^{'})_{mask} = h_g(Z^{'})_{t_3} = h_g(Z^{'})_{t_4}. \tag{15}$$

So the proof of Lemma 4 is completed. Then, we have $\text{maxpool}(h_g(Z)) = \text{maxpool}(h_g(Z^{'}))$. Theorem 1 is also proved.

Note that "$mask$" is a special token and its word embedding is a zero vector. If the hidden state before the first token is initialized as a zero vector, it can be aligned with a "$mask$" token. If a token is masked out in the input text, we can implement the operation by replacing it with a "$mask$" token as well.

## A.2  The proof of Theorem 1 when the encoder is based on Transformer

In fact, we only need to prove Lemma 4 because Theorem 1 can be easily derived from it. We express $h_g$ as :

$$h_g(X)_{x_i} = \varphi_g(e_i) + \sum_j \psi_g(e_i, e_j) \odot \varphi_g(e_j). \tag{16}$$

If $x_j$ is an uninformative token, it has no influence on $h_g(X)_{x_i}$ for any $x_i$ according to Lemma 2. So, we have $||\varphi_g(e_j)||$=0 for any uninformative $x_j$.

For any $Z_{uninf}$ that contains only uninformative tokens and $z_i, z_j$ in $Z_{uninf}$, we have

$$||h_g(Z_{uninf})_{z_i}|| = ||h_g(Z_{uninf})_{z_j}|| = 0. \tag{17}$$

The proof of Lemma 4 is completed.

# B  Details of experimental setup

## B.1  Datasets

**Beer Reviews** Following (Chang et al., 2019; Huang et al., 2021; Yu et al., 2021), we consider a classification setting by treating reviews with ratings $\leq 0.4$ as negative and $\geq 0.6$ as positive. Then we randomly select examples from the original training set to construct a balanced set.

**Hotel Reviews** Similar to Beer Reviews, we treat reviews with ratings $< 3$ as negative and $> 3$ as positive.

More details are in Table 8. $Pos$ and $Neg$ denote the number of positive and negative examples in each set. $Sparsity$ denotes the average percentage of tokens in human-annotated rationales to the whole texts.

We get the license of Beer Reviews by sending an email to Julian McAuley. The Hotel Reviews is a public dataset and we get it from https://github.com/kochsnow/distribution-matching-rationality.

## B.2  The setup of the experiment in Figure 1

We use the dataset of Beer-Aroma, whose details are shown in Table 8. We use a fixed batch size of 256 and initialize the learning rates of the two modules to different values as shown in Figure 1. The values in the cells are F1 scores that indicate the overlaps between the selected tokens and human-annotated rationales. For different settings, they use the same hyperparameters except the learning rates.

Table 8: Statistics of datasets used in this paper

| Datasets | | Train | | Dev | | Annotation | | |
|---|---|---|---|---|---|---|---|---|
| | | Pos | Neg | Pos | Neg | Pos | Neg | Sparsity |
| Beer | Appearance | 16891 | 16891 | 6628 | 2103 | 923 | 13 | 18.5 |
| | Aroma | 15169 | 15169 | 6579 | 2218 | 848 | 29 | 15.6 |
| | Palate | 13652 | 13652 | 6740 | 2000 | 785 | 20 | 12.4 |
| Hotel | Location | 7236 | 7236 | 906 | 906 | 104 | 96 | 8.5 |
| | Service | 50742 | 50742 | 6344 | 6344 | 101 | 99 | 11.5 |
| | Cleanliness | 75049 | 75049 | 9382 | 9382 | 99 | 101 | 8.9 |

## B.3 The details of skewed predictor

The experiment was first designed by Yu et al. (2021). It deliberately induces degeneration to show the robustness of A2R compared to RNP. We first pretrain the predictor separately using only the first sentence of input text, and further cooperatively train the predictor initialized with the pretrained parameters and the generator randomly initialized using normal input text. In Beer Reviews, the first sentence is usually about appearance. So, the predictor will overfit to the aspect of *Appearance*, which is uninformative for *Aroma* and *Palate*.

"*skewk*" denotes the predictor is pre-trained for $k$ epochs. To make a fair comparison, we keep the pre-training process the same as that of A2R: we use a batch size of 500 and a learning rate of 0.001.

## B.4 More visualized results

We also provide the visualized rationales from DMR and A2R corresponding to the examples in Table 3. The results are shown in Table 9.

## B.5 Visualized results of Lemma 3

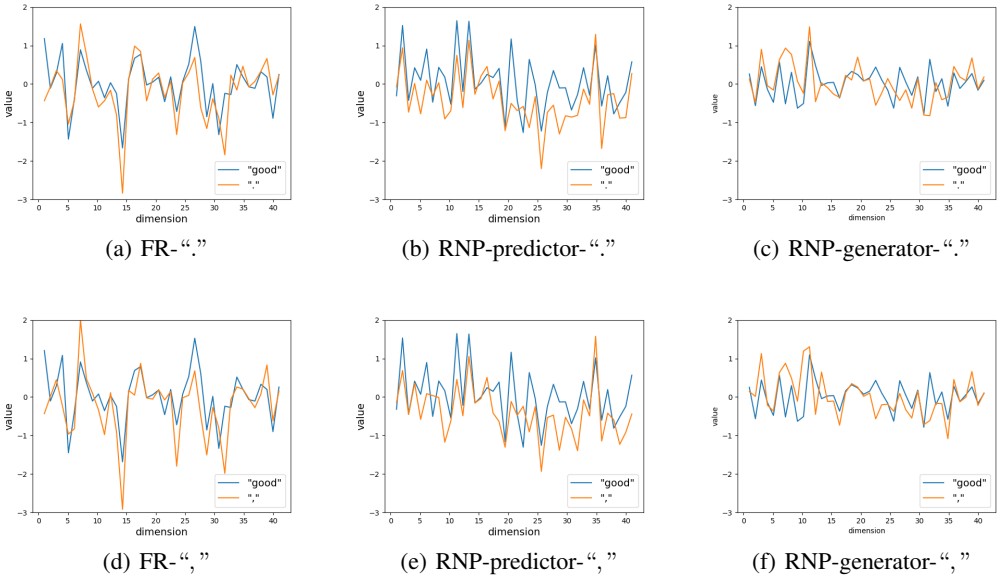

(a) FR-"."     (b) RNP-predictor-"."     (c) RNP-generator-"."

(d) FR-","     (e) RNP-predictor-","     (f) RNP-generator-","

Figure 4: Visualized examples of Lemma 3

To verify Lemma 3, we visualized the representations of the tokens in two special designed sentences "good . , smell" (a,b,c) and "good , . smell" (d,e,f) through different encoders in Figure 4. We choose these two sentences because "," and "." are uninformative in most cases, which is consistent with the setting of Lemma 3. Both our FR and RNP are trained on Beer-Aroma, where the predictive accuracy

Table 9: Examples of generated rationales. Human-annotated rationales are underlined. Rationales from RNP, DMR, A2R and FR are highlighted in red, orange, purple and blue respectively.

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

of the two models are similar while RNP selects bad rationales. Note that in FR, the encoder of the predictor is just the generator's encoder.

According to Lemma 3, the hidden states of ", " and "." should be the same as that of the word "good" after they are passed through the generator's encoder, thus resulting in the same representation. We take the first 40 dimensions for better observation. From Figure 4, we can see for both RNP and FR, the uninformative tokens ", " and "." are closer to "good" by the encoder of the generator than by the encoder of the predictor. These results directly verify the conclusion of Lemma 3.

## C   Boarder impact

The interpretability of neural networks aims to improve trust in the models. However, the problem of degeneration in the traditional two-phase rationalization models makes them untrustworthy for humans. Although the regularization terms on the predictor introduced by some follow-up work can alleviate this problem, adjusting the regularization terms requires significant manual effort to see whether the model has degenerated or not according to the human-annotated rationales, which are very costly and only present in small numbers in the test set. Our FR does not suffer from degeneration as long as the predictive accuracy is high, which can be tuned based on the validation set. So, our model saves a lot of labor cost. In addition, our model reduces the number of parameters by roughly half compared to traditional RNP and can be used by more applications with limited computational resources.

## D   Experiments with specialized layers

To make a fair comparison with the baseline models, we use the same setting and focus on the encoder with only one layer of GRUs for the experiments in the main body of our paper.

To better understand the behavior when the encoders are partly shared, we further constructed the encoders with multiple layers of GRUs as shown in Table 10 and Table 11.

Table 10: Experimental results on 3-layer encoders with different numbers of shared layers. 0 (RNP): all 3 layers are not shared. 1st: only the 1st layer is shared. 1st+2nd: the 1st and 2nd layers are shared. 3rd: only the 3rd layer is shared. 2st+3rd: the 2nd and 3rd layers are shared. all (FR): the unified encoder (all 3 layers are fully shared).

| Datasets | Beer-Aroma | | | | | Hotel-Cleanliness | | | | |
|---|---|---|---|---|---|---|---|---|---|---|
| | S | Acc | P | R | F1 | S | Acc | P | R | F1 |
| 0 (RNP) | 15.7 | 82.9 | 63.4 | 64.0 | 63.7 | 9.8 | 97.0 | 9.0 | 9.9 | 9.4 |
| 1st | 14.4 | 83.5 | 68.2 | 62.9 | 65.4 | 10.1 | 96.0 | 7.5 | 7.7 | 7.6 |
| 1st+2nd | 15.2 | 84.8 | 75.2 | 73.2 | 74.2 | 10.4 | 97.0 | 20.2 | 23.7 | 21.8 |
| 3rd | 15.5 | 82.4 | 64.2 | 63.9 | 64.1 | 9.7 | 97.0 | 32.5 | 35.7 | 34.0 |
| 2nd+3rd | 14.6 | 83.9 | 74.1 | 69.5 | 71.7 | 10.0 | 97.5 | 32.6 | 36.8 | 34.5 |
| all (FR) | 15.3 | 86.7 | 75.2 | 74.5 | 74.9 | 11.2 | 97.0 | 34.3 | 43.4 | 38.3 |

Table 11: Experimental results on 5-layer encoders with different numbers of shared layers. 0 (RNP): all layers are not shared. 1: the first layer is shared. 2: the first two layers are shared. 3: the first three layers are shared. 4: the first four layers are shared. all (FR): all layers are shared.

| Beer-Aroma | 4-layer | | | | | 5-layer | | | | |
|---|---|---|---|---|---|---|---|---|---|---|
| | S | Acc | P | R | F1 | S | Acc | P | R | F1 |
| 0 (RNP) | 14.1 | 81.2 | 73.2 | 66.6 | 69.4 | 17.1 | 85.8 | 66.9 | 75.7 | 71.0 |
| 1 | 16.7 | 85.1 | 68.0 | 73.1 | 70.5 | 14.3 | 83.6 | 74.3 | 68.4 | 71.2 |
| 2 | 14.8 | 84.7 | 72.8 | 69.1 | 70.9 | 14.7 | 85.2 | 74.2 | 70.1 | 72.1 |
| 3 | 14.9 | 82.3 | 75.5 | 72.3 | 73.9 | 15.6 | 85.6 | 71.0 | 71.0 | 71.0 |
| 4 | - | - | - | - | - | 16.3 | 87.7 | 71.9 | 75.2 | 73.5 |
| all (FR) | 15.1 | 88.0 | 76.8 | 74.3 | 75.1 | 15.4 | 87.3 | 75.2 | 74.2 | 74.7 |

It can be seen from the above tables that the case with the encoder fully shared between the generator and predictor always outperforms the cases where the encoders are partly shared in terms of rationale quality (F1 Scores). Besides, the above tables showed that the rationale quality gets better as the number of shared layers increases in overall.

We argue that, in order to obtain a good rationale, both the generator and predictor have to capture and encode the same informative sections from an input. Therefore, sharing the entire encoder can help performance. We also showed in the theoretical analysis of the paper, the encoder should be fully shared to make the predictor and the generator regularized by each other to get more stable models.