# OpenReview forum: "FR: Folded Rationalization with a Unified Encoder"
_NeurIPS.cc/2022/Conference — NeurIPS 2022 Accept_

### Official Review · Reviewer_7cgR · 2022-07-02

**Rating:** 5
**Confidence:** 4
**Soundness:** 2 fair
**Presentation:** 3 good
**Contribution:** 2 fair

**Summary:**

This paper proposes a new method for the task of rationalized classification, where a (latent, discrete) rationale is extracted and then used in a prediction module to render a classification decision. The core idea of this paper is to share the encoders for both the generator (= selector) and the predictor modules. This model is trained using a sparsity and coherence regularizer from prior work. The paper presents some theoretical results on why this rationalizer structure should be effective.  Results on two rationalized classification datasets, BeerAdvocate and Hotel Reviews, show that the proposed approach produces rationales that agree more closely with human-labeled gold standards.

**Questions:**

My main questions are around the theoretical questions above, specifically my objections about Definition 1 / Lemma 1 / Lemma 3. Please clarify these points.

**Limitations:**

The conclusion and limitations section is quite vague, with claims like "Existing works have proposed many extra modules to solve the degeneration problem, which demonstrate great effectiveness." The limitations listed here are largely conceptual things that haven't been done yet, as opposed to the (in my opinion quite real) limitations of the present technique: what datasets it can apply to, the assumptions it makes about classification, the limited scope of models considered, etc.

**Strengths And Weaknesses:**

STRENGTHS

- The problem of rationalizing predictions is an important one. Being able to do this without any supervision is a worthy goal.

- The paper compares against results from several pieces of recent prior work and shows strong improvements over these on the automatic evaluation of rationale extraction.

- The proposed method is simple and could be widely adopted for this task as a result.

WEAKNESSES

- I'm not convinced of the correctness of the theory in this work.

Let's start with the definition of uninformativeness. This is defined in terms of the conditional probability of the label given a particular token.  However, I think Definition 1 makes a very specific assumption about the structure of language that really doesn't hold.  What about the word "not"? We can imagine that "not" is uninformative by the definition here; perhaps it occurs in equal measure with positive and negative labels in the dataset. But "not good" and "not bad" might be meaningful collocations that the encoder has to account for.

I believe this case breaks Lemma 1: the behavior of the model is changed by inserting "not". Moreover, I don't see how Lemma 3 holds for cases like this with a well-trained encoder. (I don't see derivations of either lemma from first principles; these are simply asserted, and I disagree with them as they are asserted).

This state of affairs is worse for pre-trained Transformer models, where the holdover from the pre-training objective makes much of the reasoning about the objective for this particular problem moot anyway, as the encoder's behavior reflects the base language modeling objective as well.

Even setting aside collocations, I am also skeptical of Lemma 3 based on the mathematics of the models themselves. I would believe there are some idealized GRUs that this can be true for, but has this been verified on GRUs for real problems? I think the burden of proof is on the authors to show that there isn't some small latent state update (e.g., counting "uninformative" words) going on at each timestep. Otherwise, if the forget gates are nonzero even for some positions in the latent state vector, this lemma won't hold.

As a result, I don't buy the theoretical results in this paper.

- Setting aside the theory, the core contribution of this paper is pretty minor, largely an engineering one of sharing an encoder. And given that most neural modes are overparameterized, I'm not sure how much sharing an encoder actually constrains the representations that get learned even empirically. The model does seem to work better, possibly due to a multitasking effect, but again, I'm not convinced that it's for the reasons the paper states.

- The models used in this paper are very weak. 200-dimensional GRUs on 100-dimensional GloVe embeddings are simply not representative of the state-of-the-art in 2022 -- RoBERTa/ELECTRA/DeBERTa/etc. encoders should be considered. I think the empirical results about multitask learning of the predictor and generator actually depend on decisions like this, and I could imagine the outcomes being different with these models.

- The paper uses word overlap as an automatic metric but does not do any human evaluation of the produced rationales -- I could imagine that rationales from other methods may disagree with human ground truths but still illustrate the prediction in a meaningful way. I think human evaluation would strengthen the empirical claims.

---

> ### Author Response · Authors · 2022-08-03
> **Limitations and references**
>
> **Q7**. The conclusion and limitations section is quite vague, with claims like "Existing works have proposed many extra modules to solve the degeneration problem, which demonstrate great effectiveness." The limitations listed here are largely conceptual things that haven't been done yet, as opposed to the (in my opinion quite real) limitations of the present technique: what datasets it can apply to, the assumptions it makes about classification, the limited scope of models considered, etc.
>
>
> **A7**. Thank you for your valuable suggestions. The limitations may be that our theoretical analysis mainly focuses on the idealized models which have perfectly fitted the distribution of the dataset, which we have added in the revised paper.
>
>
>
>
> References
> [1] Chang, S.; Zhang, Y.; Yu, M.; and Jaakkola, T. S. 2019. A Game Theoretic Approach to Class-wise Selective Rationalization. In Advances in Neural Information Processing Systems 32: Annual Conference on Neural Information Processing Systems 2019, NeurIPS 2019, December 8-14, 2019, Vancouver, BC, Canada, 10055–10065.
> [2] Chang, S.; Zhang, Y.; Yu, M.; and Jaakkola, T. S. 2020. Invariant Rationalization. In Proceedings of the 37th International Conference on Machine Learning, ICML 2020, 13-18 July 2020, Virtual Event, volume 119 of Proceedings of Machine Learning Research, 1448–1458. PMLR.
> [3] Huang, Y.; Chen, Y.; Du, Y.; and Yang, Z. 2021. Distribution Matching for Rationalization. In Thirty-Fifth AAAI Conference on Artificial Intelligence, AAAI 2021, Thirty-Third Conference on Innovative Applications of Artificial Intelligence, IAAI 2021, The Eleventh Symposium on Educational Advances in Artificial Intelligence, EAAI 2021, Virtual Event, February 2-9, 2021, 13090–13097. AAAI Press.
> [4] Yu, M.; Chang, S.; Zhang, Y.; and Jaakkola, T. S. 2019. Rethinking Cooperative Rationalization: Introspective Extraction and Complement Control. In Proceedings of the 2019 Conference on Empirical Methods in Natural Language Processing and the 9th International Joint Conference on Natural Language Processing, EMNLP-IJCNLP 2019, Hong Kong, China, November 3-7, 2019, 4092–4101. Association for Computational Linguistics.
> [5] Yu, M.; Zhang, Y.; Chang, S.; and Jaakkola, T. S. 2021. Understanding Interlocking Dynamics of Cooperative Rationalization. In Advances in Neural Information Processing Systems 34: Annual Conference on Neural Information Processing Systems 2021, NeurIPS 2021, December 6-14, 2021, virtual, 12822–12835.
> [6] Sha, L.; Camburu, O.; and Lukasiewicz, T. 2021. Learning from the Best: Rationalizing Predictions by Adversarial Information Calibration. In Thirty-Fifth AAAI Conference on Artificial Intelligence, AAAI 2021, Thirty-Third Conference on Innovative Applications of Artificial Intelligence, IAAI 2021, The Eleventh Symposium on Educational Advances in Artificial Intelligence, EAAI 2021, Virtual Event, February 2-9, 2021, 13771–13779. AAAI Press.
> [7] Shen, H.; Wu, T.; Guo, W.; and Huang, T. K. 2022. Are Shortest Rationales the Best Explanations for Human Understanding? In Proceedings of the 60th Annual Meeting of the Association for Computational Linguistics (Volume 2: Short Papers), ACL 2022, Dublin, Ireland, May 22-27, 2022, 10–19. Association for Computational Linguistics.
> [8] Chan, A.; Sanjabi, M.; Mathias, L.; Tan, L.; Nie, S.; Peng, X.; Ren, X.; and Firooz, H. 2022. UNIREX: A Unified Learning Framework for Language Model Rationale Extraction. In International Conference on Machine Learning, ICML 2022, 17-23 July 2022, Baltimore, Maryland, USA, volume 162 of Proceedings of Machine Learning Research, 2867–2889. PMLR.
> [9] Chen, H.; He, J.; Narasimhan, K.; and Chen, D. 2022. Can Rationalization Improve Robustness? In Carpuat, M.; de Marneffe, M.; and Ru ́ız, I. V. M., eds., Proceedings of the 2022 Conference of the North American Chapter of the Association for Computational Linguistics: Human Language Technologies, NAACL 2022, Seattle, WA, United States, July10-15, 2022, 3792–3805. Association for Computational
> Linguistic.
> [10] Guerreiro, N. M.; and Martins, A. F. T. 2021. SPECTRA: Sparse Structured Text Rationalization. In Proceedings of the 2021 Conference on Empirical Methods in Natural Language Processing, EMNLP 2021, Virtual Event / Punta Cana, Dominican Republic, 7-11 November, 2021, 6534–6550. Association for Computational Linguistics.
> [11] Paranjape, B.; Joshi, M.; Thickstun, J.; Hajishirzi, H.; and Zettlemoyer, L. 2020. An Information Bottleneck Approach for Controlling Conciseness in Rationale Extraction. In Proceedings of the 2020 Conference on Empirical Methods in Natural Language Processing, EMNLP 2020, Online, November 16-20, 2020, 1938–1952. Association for Computational Linguistics.

---

> ### Author Response · Authors · 2022-08-03
> **Discussion about Bert and metrics**
>
> **Q5**. The models used in this paper are very weak.
> 200-dimensional GRUs on 100-dimensional GloVe embeddings are simply not representative
> of the state-of-the-art in 2022 -- RoBERTa/ELECTRA/DeBERTa/etc. encoders should be considered.
> I think the empirical results about multitask learning of the predictor and generator actually depend on
> decisions like this, and I could imagine the outcomes being different with these models.
>
> **A5**. Thank you for your suggestions. In the field of rationalization, researchers generally focus on frameworks of the models and the methodology. Methods most related to our work do not use Bert or other pre-trained encoders [1][2][3][4][5][6]. As a consequence, we use GRUs and GloVe to ensure the same experimental setup as our baselines for a fair comparison.
>
> More importantly, how to finetune large models on the rationalization framework is still a significant challenge.
> Some recent studies [9] show that the methods with Bert encoders perform much worse than those with simple GRUs on BeerAdvocate and HotelReviews. To verify this, we here also conduct experiments using Bert. The results are presented as follows.
>
> |          | Beer-Appearance | Hotel-Cleanliness |
> |----------|-----------------|-------------------|
> | VIB      | 20.5            | 23.5              |
> | SPECTRA  | 28.6            | 19.5              |
> | FR(ours) | 29.8            | 24.0              |
> | RNP      | 14.7            | 12.4              |
>
> VIB [11] and SPECTRA [10] are two improved rationalization methods used in [9]. The results of VIB and SPECTRA are from [9].
> We fine-tune FR and RNP for 10 epochs with a batch-size 16.
> The learning rates of the Bert encoder and the Linear layer are 2e-5 and 1e-4 respectively.
> The results show that our method is still much better than our direct baseline RNP.
>
>
> **Q6**. The paper uses word overlap as an automatic metric
> but does not do any human evaluation of the produced rationales --
> I could imagine that rationales from other methods may disagree with
> human ground truths but still illustrate the prediction in a meaningful way.
> I think human evaluation would strengthen the empirical claims.
>
> **A6**. Thank you for your suggestion.
> Human evaluation may be good to better show strength and weakness from different perspectives,
> but it will be very expensive, which may be similar to the cost of re-annotating human rationales on these datasets.
>
> To the best of our knowledge, there is only very limited research on rationalization which ever used human evaluation [7][8]. [7] uses human study to show the influence of different
> rationale lengths, while we set the average length of rationales for different baselines to a similar level
> . We have tried different
> length settings: similar to human-annotated rationales (table 2) and extremely low (table 4). [8] uses
> human study because the model’s predicted label may be different with the target label, while we focus
> on the problem of degeneration which means right prediction with bad rationales, and the predictive
> accuracies are relatively high and similar across baselines.
> Most researches including all of our baselines do not use human evaluation. The overlap with human-annotated rationales is the most widely used way to show the alignment with human in the literature of rationalization. Our metrics are the same as those of used in
> our baselines DMR and A2R [3][4].

---

> > ### Comment · Reviewer_7cgR · 2022-08-08
> > **BERT etc.**
> >
> > What metric is reported in the table?
> >
> > In [9], FC (a BERT model) = VIB on Beer and loses to SPECTRA, and beats both on Hotel. (I'm looking at "Ori" rather than the attack performance.) However, there are much better models than BERT -- ELECTRA and DeBERTa. This paper advocates for using updated comparisons like this: https://arxiv.org/pdf/2110.08300.pdf
> >
> > Human evaluation: I recognize this evaluation may be standard in other papers. But I think rationalized predictions and the field of interpretability more broadly have moved on substantially since the early work on rationalization like Lei et al., 2016. Papers like this one https://arxiv.org/pdf/2201.11569.pdf really argue that current evaluation paradigms are not sufficient. I think this kind of justification "by induction" ("this paper is okay because it improves on last year's results") is convincing for a time but eventually a field needs to progress.

---

> > > ### Author Response · Authors · 2022-08-08
> > > **Results of using different pre-trained models**
> > >
> > > We would like to express our great appreciation to you for investing the time to evaluate our work and provide such thorough feedback.
> > >
> > > We are sorry that we didn’t clearly point out the evaluation metric used in the table in our previous response. The metric is the F1 score of generating rationales, which is the same one used in our original paper. Your comments on the experimental results in [9] seem to refer to Table 3, which shows the classification accuracy of vanilla BERT. Our work focuses on rationales selected by the generator of self-explaining models, and the experimental results of [9] mentioned in our previous response actually refer to “GR” (gold rationale) in Table 4, which indicates the degree of coincidence with the manually marked rationales in terms of F1 score.
> > > Besides, we also conduct some new experiments on other pre-trained models, e.g., ELECTRA and DeBERTa. The results of F1 score on Beer-Aroma are presented in the following table.
> > >
> > >
> > > | Method/Model 	|   ELECTRA  | DeBERTa  |
> > > | ---------- | ---- | ---- |
> > > | FR    	   | 14.6 | 17.6 |
> > > | RNP          | 13.7 | 16.6 |
> > >
> > > The learning rates adopted for encoders in both experiments are $2e{-5}$, and for linear layers are $1e{-4}$. The batch-size used in ELECTRA is 32, and in DeBERTa is 4. We fine-tune ELECTRA with 40 epochs and DeBERTa with 4 epochs. Due to the limits of time and GPU resources, we do not carefully tune the hyper-parameters in a grid-search way. However, the results may still shed some light on how the pre-trained models work on rationalization, which performs much worse than GRUs. We will discuss these results in our revised paper and further study these problems by following [1].
> > >
> > > [1] Bowman, Samuel. "The Dangers of Underclaiming: Reasons for Caution When Reporting How NLP Systems Fail." Proceedings of the 60th Annual Meeting of the Association for Computational Linguistics (Volume 1: Long Papers). 2022.
> > >
> > > We agree that human evaluation is more convincing for rationalization. However, bringing human evaluation into this field is still challenging, such as selecting participants for a fair comparison. We will definitely take human evaluation into account in the future.

---

> ### Author Response · Authors · 2022-08-03
> **Contributions and relationship with multi-task learning**
>
> **Q4**. Setting aside the theory, the core contribution of this paper is pretty minor,
> largely an engineering one of sharing an encoder.
> And given that most neural modes are overparameterized, I'm not sure how much sharing
> an encoder actually constrains the representations that get learned even empirically.
> The model does seem to work better, possibly due to a multitasking effect, but again, I'm not convinced that it's for the reasons the paper states.
>
> **A4**.
> Although the method of sharing encoder proposed in this paper is simple and may have been thoroughly studied in other fields, it is explored in the field of rationalization by us for the first time, which we believe has assignable novelty. Specifically, our method bridges the rationalization with those methods that share parameters or encoders. Besides, we get some motivated observations for demystifying the degeneration problem, as shown in Figure 1 of the paper. Our observations find that one of the main causes for degeneration is the imbalance of learning speed between the generator and predictor for the first time, which may motivate future methods specially designed for achieveing better balance, such as adaptively tuning learning rates for the generator and predictor separately.
>
>
> For the concerns about overparameterized models and multi-task learning, we performed more experiments where the encoders inlcude 3 times more parameters and are partly shared between the generator and decoder. The results are presented as follows.
>
> | Beer-Aroma | s    | acc  | p    | r    | f1    |
> | ---------- | ---- | ---- | ---- | ---- | ---- |
> | all(FR)    | 15.3 | 86.7 | 75.2 | 74.5 | 74.9 |
> | 1          | 14.4 | 83.5 | 68.2 | 62.9 | 65.4 |
> | 1+2        | 15.2 | 84.8 | 75.2 | 73.2 | 74.2 |
> | 3          | 15.5 | 82.4 | 64.2 | 63.9 | 64.1 |
> | 2+3        | 14.6 | 83.9 | 74.1 | 69.5 | 71.7 |
> | 0(RNP)     | 15.7 | 82.9 | 63.4 | 64.0   | 63.7 |
>
> | Hotel-Cleanliness | s    | acc  | p    | r    | f1    |
> | ----------------- | ---- | ---- | ---- | ---- | ---- |
> | all(FR)           | 11.2 | 97.0   | 34.3 | 43.4 | 38.3 |
> | 1                 | 10.1 | 96.0   | 7.5  | 7.7  | 7.6  |
> | 1+2               | 10.4 | 97.0   | 20.2 | 23.7 | 21.8 |
> | 3                 | 9.7  | 97.0   | 32.5 | 35.7 | 34.0   |
> | 2+3               | 10.0   | 97.5 | 32.6 | 36.8 | 34.5 |
> | 0(RNP)            | 9.8  | 97.0   | 9.0    | 9.9  | 9.4  |
>
> Notes:
>
> All (FR): the unified encoder (all 3 layers are fully shared)
>
> 1: only the 1st layer is shared
>
> 1+2: the 1st and 2nd layers are shared
>
> 3: only the 3rd layer is shared
>
> 2+3: the 2nd and 3rd layers are shared
>
> 0(RNP): all 3 layers are not shared
>
>
>
> The results show that our method still works effectively when the model is much larger. Besides, it can be seen from the table that the case fully sharing the encoder between the generator and predictor always outperforms that where the encoders are partly shared in terms of rationale quality (f1). Furthermore, the rationale quality gets better as the number of shared layers increases in overall, which is different
> from the common intuitation of multi-task learning that usually needs some specialized layers for each task.
> As a consequence, our method is not strongly related to the multi-task learning clearly. In fact, the theoretical principles of multi-task learning methods still remain mysterious, which are usually intuitively motivated, while we provide theoretical guarantees to our method. We appreciate your interest in the relationship between our method and multi-task learning, for which we seek to make further explorations as specified in lines 315-320.

---

> ### Author Response · Authors · 2022-08-03
> **Derivation of Lemma 3 and practical experimental verification**
>
> **Q2**. I don't see how Lemma 3 holds for cases like this with a well-trained
> encoder (I don't see derivations of either lemma from first principles; these are simply asserted, and I disagree with them as they are asserted).
> This state of affairs is worse for pre-trained Transformer models, where the holdover from the pre-training objective makes much of the reasoning about the objective for this particular problem moot anyway, as the encoder's behavior reflects the base language modeling objective as well.
>
> **A2**. Thanks for raising the concerns. For all the concerns about Lemma 3, we think that here is a major misunderstanding about our theoretical analysis. Note that our theory is not to analyze the intermediate states of the model, e.g., a not converged model in the training process or a pre-trained model going to be finetuned, but the properties of the model after it has got the optimal solution for the loss function (refer to equation 2 or previous response) as specified in lines 181-182 and lines 196-197.
>
> The derivation process in our original paper is simplified to some extent, and thus it may lose some readability. For ease of understanding the derivations of Lemma 3 from Lemma 2 and Lemma 1, we here further present the reasoning process in detail.
>
> From Lemma 2, we know that the representation of each token obtained by the generator’s encoder keeps unchanged after an uninformative token is added to the original text, under the condition that $\epsilon = 0$ in Lemma 1. For simplicity, we here consider the general one-way RNN network (the derivation process for Transformer can be found in the original Appendix.A.2). For the $i$-{th} token in $X$, its representation can be expressed as:
> \begin{equation}
>     h_g (X)~{x_i}=\varphi_g (s_{i}), \ s_{i}=\psi_g (e_i,s_{i-1}),
> \end{equation}
> where $e_i, s_i$ are the word embedding and hidden state of the $i$-{th} token, respectively. $\varphi_g,\psi_g$ are two functions corresponding to the type of RNN units.
> When an uninformative token $t$ is added between $x_i$ and $x_{i-1}$, we have
> \begin{equation}
>     s_t=\psi_g (e_t,s_{i-1}), \ s_i^{'}=\psi_g (e_i,s_{t}),
> \end{equation}
> where $s_i^{'}$ is the hidden state of the original $x_i$ in this new text and we denote it by $h_g ([X,t]) ~{x_i^{'}}=\varphi_g (s_{i}^{'})$. From Lemma 2, we have $h_g (X) ~{x_i}=h_g ([X,t]) ~{x_i^{'}}$. Using the first term of of the first equation (equation 8 in our original paper), we get
> \begin{equation}
>     \varphi_g(s_{i})=h_g (X) ~{x_i}=h_g ([X,t]) ~{x_i^{'}}=\varphi_g (s_{i}^{'}).
> \end{equation}
> Then, we have $s_i=s_i^{'}$. Using the second term of of the second equation (equation 9 in our original paper), we get
> \begin{equation}
>     \psi_g (e_i,s_{t})=s_i^{'}=s_{i}=\psi_g (e_i,s_{i-1}).
> \end{equation}
> Then, we have $s_{t}=s_{i-1}$ very smoothly and Lemma 3 is just a formal description of this equation.
>
> Here are the responses to the concerns about pre-trained models and Transformers. As discussed above, our theoretical analysis does not focus on the intermediate states of the model, but the well-trained model on the defined rationalization loss function. Therefore, how to initialize model does not affect the theoretical properties of the final model. Besides, we also did theoretical analysis for Transformer encoders, as shown in Appendix A.2. Furthermore, we conducted extensive experiments Furthermore, we conducted extensive experiments where we pre-train the generator and the predictor with special loss functions to deliberately induce degeneration. The results can be found in Table 5 and Table 6, which shows that our model still works effectively when the encoder is pre-trained.
>
>
> **Q3**. Even setting aside collocations, I am also skeptical of Lemma 3 based on the mathematics of the models themselves. I would believe there are some idealized GRUs that this can be true for, but has this been verified on GRUs for real problems? I think the burden of proof is on the authors to show that there isn't some small latent state update (e.g., counting "uninformative" words) going on at each timestep. Otherwise, if the forget
> gates are nonzero even for some positions in the latent state vector, this lemma won't hold.
>
> **A3**. Thank you for your concerns. To verify Lemma 3 for GRUs on real problems, we give some examples of the direct visualization of Lemma 3 in the course of the experiments. The result figures are uploaded to the new Appendix B.5. As discussed above, our Lemma 3 holds under the condition where the learning error of the neural network is 0, i.e., $\epsilon$=0 in Lemma 1 (lines 194-196). Please refer to the response A2 for more discussion.

---

> > ### Comment · Reviewer_7cgR · 2022-08-07
> > **Lemma 3 etc.**
> >
> > I understand that the theoretical result is about the converged model. Rereading this section now with your additional explanation is clearer. The presentation could be clearer in terms of what is considered to be a proof of each lemma or informal arguments -- right now it's hard to follow the argumentation in the text itself.
> >
> > The experimental results are also quite helpful in justifying the correctness of this lemma to me. I still feel like there's something missing that could be a counterexample (e.g., adding a vector in the nullspace of the ensuing linear function, if it isn't full-rank), but I can't come up with it.
> >
> > I've raised my score by 2 points.

---

> > > ### Author Response · Authors · 2022-08-09
> > > **Revision of paper for Lemma 3**
> > >
> > > Thank you very much for the valuable feedback and perspective. We are encouraged by the positive feedback, and we think the updates based on your helpful suggestions make the paper much stronger as a result. We will revise the paper to further clearly explain this lemma with the results added in Appendix A.3 and B.5.

---

> ### Author Response · Authors · 2022-08-03
> **Response to Definition 1 and Lemma 1**
>
> **Q1**: Let's start with the definition of uninformativeness. This is defined in terms of the conditional probability of the label given a particular token. However, I think Definition 1 makes a very specific assumption about the structure of language that really doesn't hold. What about the word "not"? We can imagine that "not" is uninformative by the definition here; perhaps it occurs in equal measure with positive and negative labels in the dataset. But "not good" and "not bad" might be meaningful collocations that the encoder has to account for.
> I believe this case breaks Lemma 1: the behavior of the model is changed by inserting "not".
>
> **A1**: Thank you very much for this comment which is greatly valuable for improving the rigor of our theoretical analysis. We agree with your point that Definition 1 described in the original paper is not suitable and Lemma 1 cannot be derived from the original Definition 1. In fact, this misleading error arises from the inaccurate description we made for Definition 1. More specifically, what we sought to define was a conditional independent distribution among the text $X$, the uninformative token $t$, and the category $Y$. To clarify the misunderstanding above, we revise the description as follows.
>
> **Revised Definition 1**
> For $X \sim \mathcal{D}$, $X_s$ is a subset of $X$, for a given token $t$, we say it is uninformative for $X$ if the following equation holds:
> \begin{equation}
>      \mathbb{P}(Y|X_s)=\mathbb{P}(Y|X_s,t),\quad s.t.,\ \forall X_s \subset X.
> \end{equation}
>
> The revised definition says that the token $t$ is independent of $Y$ under the condition of $X_s$, where $t$ is defined to be uninformative to $Y$ for $X$. In our original paper, Lemma 1 is derived from this revised definition which we should have given strictly. To verify this statement, we further present the derivative process of obtaining Lemma 1 from the revised definition.
>
> We consider that equation 2 of the original paper (loss function, $\mathop{\min}\limits_{\theta_g,\theta_p}H_{(X,Y) \sim \mathcal{D}}(Y,pred(gen(X)))$) gets the optimal solution where the model perfectly fits the data distribution. Then, we have $pred(gen(X))=  \mathbb{P}(Y|X)$ and $pred(gen([X,t]))= \mathbb{P}(Y|X,t)$, where "[X,t]" denotes the result of adding token $t$ to any position of $X$. According to Deinition 1, we know that $\mathbb{P}(Y|X)= \mathbb{P}(Y|X,t)$ holds for any uninformative $t$. As a consequence, we can immediately get that $pred(gen(X)) = \mathbb{P}(Y|X)=\mathbb{P}(Y|X,t) =pred(gen([X,t]))$, which proves Lemma 1.
>
> For ease of reading, we here further present Lemma 1.
> **Lemma 1**
> When an uninformative token $t$ is added to the original text $X$, the predictive results will hardly change:
> \begin{equation}
>     pred(gen(X)) = pred(gen([X,t]))+\epsilon, \ s.t.,\ \forall X_s \subset X, \ \mathbb{P}(Y|X_s)=\mathbb{P}(Y|X_s,t),
> \end{equation}
> where $\epsilon$ is an error depending on the learning error of the neural network. When the model gets the optimal solution, the absolute value of $\epsilon$ is $0$.
>
> We thank again for the insightful comment.

---

> > ### Comment · Reviewer_7cgR · 2022-08-07
> > **Defn 1**
> >
> > Thanks for the update. Revised Definition 1 is a pretty major change in the definition of uninformativeness. This is a much stronger criterion, and it's a bit unclear to me how true this is in practice. Y has to be independent of the token given *every* subset of the input? Even an article like "the" possibly has some correlation with some meaningful factor in some subset (e.g., in a short review, having a definite noun phrase might indicate a specific comment which is positive). In any case, I recognize that with this definition, the other math makes a lot more sense.

---

> > > ### Author Response · Authors · 2022-08-08
> > > **More discussion about Definition 1**
> > >
> > > Thank you very much again for your suggestions that refine our definition. In fact, our definition comes from the implicit assumption of rationalization, which is "any unselected input is guaranteed to have no contribution to prediction" [5]. We in this paper simply quantify this assumption in a formal way. Besides, we quantify the uninformativeness (i.e., non-contribution) of the token which is calculated by every subset of the input, by referring to the concept of the Shapley value which is generally adopted to indicate how much contribution an element makes in a given set.
> > >
> > >
> > > Indeed, every word may be meaningful semantically. However, rationalization defines the informativeness or uninformativeness of a word under some specific task. That is to say, a word is uninformative when it is uncorrelated with the given specific task. To understand this, we can consider the following example.
> > >
> > >
> > > Hotel-Cleanliness
> > > **Label**: positive, **Prediction**: positive.
> > > **Text** :  my husband and i just spent two nights at the grand hotel francais , and we could not have been happier with our choice . in many ways , $\underline{\text{the hotel has exceeded our expectation}}$ : $\color{red}{\text{the price was within our budget , breakfast was included,}}$ and the staff was friendly , helpful and fluent in english. as other travelers have mentioned , the hotel is close to the nation metro station , which makes it easy to get around . the room size was just enough to fit two people , but $\underline{\text{we had a comfortable stay throughout }}$. overall , the hotel lives up to its high trip advisor rating . we would love to stay here again anytime .
> > >
> > > Text in red: an example of uninformative texts.
> > > Text underlined: the rationales annotated by human.
> > >
> > > The task is to judge whether the hotel is clean. It is apparent that the text in red contributes nothing to the task, whenever it co-exists with the other texts.

---

> > > > ### Comment · Reviewer_7cgR · 2022-08-08
> > > > **Example doesn't agree with the definition?**
> > > >
> > > > t is uninformative if it doesn't change the probability *for every subset X_s*, right?
> > > >
> > > > Let's take the word "included" highlighted in red. Set X_s = "breakfast was". If I see "breakfast was included" as the whole review, this may be a better and nicer hotel and more likely to be clean. So P(Y | breakfast was) < P(Y | breakfast was included). So by that measure, "included" cannot be an uninformative token, right?
> > > >
> > > > I agree that P(Y | black text) = P(Y | black text, red text), but that's not what Definition 1 is saying.

---

> > > > > ### Author Response · Authors · 2022-08-09
> > > > > **Discussion about Definition 1**
> > > > >
> > > > > Thank you for your responses.
> > > > > In fact, both "breakfast was" and "breakfast was included" mainly discuss the service instead of the cleanliness of the hotel. Chances are that there is extra service for fees with bad cleanliness. Therefore, service and cleanliness should be regarded as two independent events from the perspective of causal understanding of humans. Furthermore, we can understand this case from the view of Bayesian, which has also been thoroughly discussed in Figure 2 of [2]. Recently, some research has tried to solve the problem of spurious correlation between different texts[1][2].
> > > > >
> > > > >
> > > > > We agree with you that the case considered may be idealized. To further clarify why our definition is reasonable, we have to agree on two intuitions that are common in rationalization and XAI, respectively.
> > > > > **Intuition 1**  Uninformative texts have no contribution to the prediction.
> > > > > **Intuition 2** Shapley value is a good metric to estimate the contribution of a subset.
> > > > > Under these two intuitions, we argue that our definition is reasonable because it is just how the Shapley value is calculated for no contribution, i.e., all marginal contributions are zero.
> > > > >
> > > > > The reason why it seems to be too strong in practice is that Intuition 1 is idealized in the field of binary selection. Ideally, the parts not regarded as explanations should not have any contribution to the prediction. If the length of the explanation is not constrained, all components that contribute to the prediction should be considered as part of the explanation. However, as compared to a saliency map in which every part contributes more or less, rationalization gets the explanation through binarized selection where the unselected parts are considered as no contribution.
> > > > >
> > > > >
> > > > > In practice, we agree that it is hard to say unselected parts have no contribution at all because every part of the input may contribute more or less. In rationalization, there is a trade-off between Intuition 1 and rationale length, controlled by $\lambda_1$ of the sparsity regularizer in equation 3. Then, Intuitation 1 may be modified as uninformative texts make very little contribution to the prediction. In this way, another small error $\epsilon$ should be added to the equation of Definition 1. Although our analysis is under the original Intuition 1 without modification, we argue that the theoretical conclusion will keep almost the same but with some tiny errors. Anyway, the condition "for every subset $X_s$" makes great sense because the Shapley value is still the most widely accepted metric for estimating the contribution in XAI. In the future, we will seek to identify better metrics for practice.
> > > > >
> > > > > Although the case we analyze is ideal to some extent, we are the first to formally analyze how uninformative texts influence the rationalization model. We believe that the established theory still makes great sense by providing motivations for future better works.
> > > > >
> > > > > Finally, whatever the outcome is, we appreciate it very much that you have joined the discussion with us and provided many valuable suggestions for improving the quality of our paper.
> > > > >
> > > > > [1]Plyler, M.; Green, M.; and Chi, M. 2021. Making a (Coun-
> > > > > terfactual) Difference One Rationale at a Time. In Ad-
> > > > > vances in Neural Information Processing Systems 34: An-
> > > > > nual Conference on Neural Information Processing Systems
> > > > > 2021, NeurIPS 2021, December 6-14, 2021, virtual, 28701–
> > > > > 28713
> > > > > [2]Chang, S.; Zhang, Y.; Yu, M.; and Jaakkola, T. S. 2020. In-
> > > > > variant Rationalization. In Proceedings of the 37th Interna-
> > > > > tional Conference on Machine Learning, ICML 2020, 13-18
> > > > > July 2020, Virtual Event, volume 119 of Proceedings of Ma-
> > > > > chine Learning Research, 1448–1458. PMLR

---

> > > > > > ### Comment · Reviewer_7cgR · 2022-08-09
> > > > > > **Shapley values**
> > > > > >
> > > > > > The problem I'm pointing out is discussed in Section 3.1 of this paper:
> > > > > >
> > > > > > Kumar et al. "Problems with Shapley-value-based explanations as feature importance measures"
> > > > > > https://www.semanticscholar.org/reader/364f02eff4f10ea602d86fd8c98c8694b76f46fd
> > > > > >
> > > > > > I think you're trying to make an interventional argument from conditional distributions. "Therefore, service and cleanliness should be regarded as two independent events from the perspective of causal understanding of humans" can be true, while the correlation I pointed out may also be true. Anyway, this discussion has arrived at very fundamental limitations of this type of rationalization work which are a bit outside the scope of this paper.
> > > > > >
> > > > > > Thanks for the thorough discussion about this paper! This has been illustrative for me and clarified my thoughts about the work significantly.

---

> > > > > > > ### Author Response · Authors · 2022-08-09
> > > > > > > **Thank you again and whether all the concerns have been addressed?**
> > > > > > >
> > > > > > > It is our duty to clarify what is not clear. Thank you very much again for the extensive comments you gave and the helpful references that we did not cover. Your thorough reading and concerns play an important role in making our work more comprehensive and complete. I think the process of discussion with you are really enjoyable. By the way, we hope that all your concerns have been addressed well. If you have any other concerns, we will be glad to talk with you further.
> > > > > > >
> > > > > > > Thank you again for spending time discussing with us! With best wishes to you and yours!

---

### Official Review · Reviewer_iQBE · 2022-07-11

**Rating:** 7
**Confidence:** 2
**Soundness:** 3 good
**Presentation:** 3 good
**Contribution:** 3 good

**Summary:**

This paper investigates the rationalization for NLP models. The authors analyze the relationship between generator and predictor in terms of working mechanism and propose folded rationalization that folds two phases of the rationalization into one via a unified encoder. Results show that FR improves over the existing baselines on beer and hotel review datasets.

**Questions:**

N/A

**Limitations:**

Yes

**Strengths And Weaknesses:**

Pros:
-	The proposed approach is simple but effective for rationalization in NLP.
-	The authors conduct in-depth analysis to show that the predictor in FR is regularized by the generator through the unified encoder.
-	Experiments show that the proposed approach outperforms baselines by a large margin.


Cons:
-	All experiments are conducted on two review datasets. It would be better to show the effectiveness of FR on other tasks.

---

> ### Author Response · Authors · 2022-08-03
> **Discussion about the datasets**
>
> Q1. All experiments are conducted on two review datasets. It would be better to show the effectiveness of FR on other tasks.
>
>
> A1. Thanks for your suggestion, and in the future we definitely will try to apply our method to some other datasets and tasks.
>
>
>
> We selected the current two datasets for the experiments, because most current elf-explanatory models via rationalization such as CAR[1] and DMR [2] used the same datasets as the benchmark, which is the common settings we followed in the field. Each of the two datasets contains 3 aspects and every aspect is trained independently. They can be viewed as 6 individual datasets, which represent typical scenarios of sentimental analysis.
>
>
>
> Reference
> [1] Chang, S.; Zhang, Y.; Yu, M.; and Jaakkola, T. S. 2019. A Game Theoretic Approach to Class-wise Selective Rationalization. In Advances in Neural Information Processing Systems 32: Annual Conference on Neural Information Processing Systems 2019, NeurIPS 2019, December 8-14, 2019, Vancouver, BC, Canada.
> [2] Yongfeng Huang, Yujun Chen, Yulun Du, and Zhilin Yang. Distribution matching for ratio340 nalization. In Proceedings of the AAAI Conference on Artificial Intelligence, 2021.

---

> > ### Author Response · Authors · 2022-08-09
> > **Thanks for reviewing the paper**
> >
> > Thank you very much for taking the time to review our paper! With best wishes to you and yours!

---

### Official Review · Reviewer_oaM2 · 2022-07-12

**Rating:** 6
**Confidence:** 3
**Soundness:** 3 good
**Presentation:** 3 good
**Contribution:** 3 good

**Summary:**

This paper addresses the degeneration problem occurring in the two-phase models for rationalization. The authors identified that the learning speeds of predictor and generator are out of sync. Thus the predictor overfits to the uninformative pieces selected by the not yet well-trained generator, leading to a sub-optimal model. The authors propose FR that uses a shared encoder to fold the two phases into one such that the learning speeds of two components can be aligned. The authors’ claims are supported empirically and theoretically.

**Questions:**

Have you considered adding specialized layers in each component’s encoder such that the encoders are partly shared but not identical?

It may be worthy if you can present some failure cases where the model failed to select reasonable rationales. This can pave ways to gain better insights of the model and be helpful to this line of research.


**Limitations:**

Yes, the paper discusses the limitations of their work.

**Strengths And Weaknesses:**

Strengths:

(1) The observation and the idea presented in the paper is interesting. Folding two phases of training into one enforces that the learning processes of two components are compatible, addressing the degeneration problem.
(2) The claims are well supported by both theoretical analysis and solid empirical results.
(3) The paper is generally well written and easy to follow. The idea and motivation are presented clearly.
(4) The topic discussed in the paper is valuable to the broader community, as it aims to improve the interpretability of NLP models.

Weaknesses:

The paper did not thoroughly discuss the discrepancy when using a unified encoder. The functions of two components are related but not identical. While parts of the encoder can be shared indeed, I cannot see why the unified encoder can work well if there are no other specialized layers in each component’s encoder.

---

> ### Author Response · Authors · 2022-08-03
> **Part2-Discussion about failure cases**
>
> Q2: It may be worthy if you can present some failure cases where the model failed to select reasonable rationales. This can pave ways to
> gain better insights of the model and be helpful to this line of research.
>
> A2: We have added the failure cases to the updated version of our Appendix B.6. For convenience of analysis, we further list some cases as follows.
>
> Beer-Palate
> **Label**: positive, **Prediction**: negative.
> **Text**:  pours a slight tangerine orange and straw yellow . the head is nice and bubbly but fades very quickly with a little lacing . smells like wheat and european hops , a little yeast in there too . there is some fruit in there too , but you have to take a good whiff to get it the taste is of wheat , a bit of malt , and a $\color{blue}{\text{little} }$ fruit $\color{blue}{\text{flavour\ in\ there\ too}}$ $\underline{\text{\color{blue}{almost}}\  \text {feels like drinking champagne , medium mouthful otherwise}}$ easy to drink , but not somthinf i 'd be trying every night
>
>
> Hotel-Cleanliness
> **Label**: positive, **Prediction**: positive.
> **Text** :  my husband and i just spent two nights at the grand hotel francais , and we could not have been happier with our choice . in many ways , $\underline{\text{the hotel has exceeded our expectation}}$ : the price was within our budget , $\color{blue}{\text{breakfast}}$ was included , $\color{blue}{\text{and the staff was friendly , helpful and}}$ fluent in english . as other travelers have mentioned , the hotel is close to the nation metro station , which makes it easy to get around . the room size was just enough to fit two people , but $\underline{\text{we had a comfortable \color{blue}{stay\ } throughout }}$. overall , the hotel lives up to its high trip advisor rating . we would love to stay here again anytime .
>
> Beer-Aroma
> **Label**: positive, **Prediction**: positive.
> **Text**: a- amber gold with a solid two maybe even three finger head . looks absolutely $\color{blue}{\text{delicious , i dare say it is one of the best looking beers i 've had}}$ . $\underline{\text{s- light citrus and hops . not a very strong \color{blue}{aroma}}}$ t-wow , $\color{blue}{\text{the}}$ hops , $\color{blue}{\text{citrus}}$ and $\color{blue}{\text{pine}}$ blow out the taste buds , very tangy in taste , yet perfectly balanced , leaving a crisp dry taste to the palate . m-light and crisp feel with a nice tanginess thrown in the mix . d- could drink this all night , too bad i only have one more of this brew . notes : one of the best balanced and best tasting ipa 's i 've had to date . ipa fans you have to try this one .
>
> Text with blue: the rationales selected by our generator.
> Text with underline: the rationales annotated by human.
>
> Typically, the failures can be summarized as follows. First, the model fails to grasp high-level language phenomena and commonsense. For example, in the case of Beer-Palate, the model fails to understand the analogy statement of "almost feels like drinking champagne" and makes the prediction as negative.
> Second, the model fails to make logical reasoning. The ground truth of hotel cleanliness should be inferred from the word "comfortable". However, our model makes judgments by selecting the rationales such as "breakfast, staff was friendly..., stay", which have little relevance to the cleanliness but are positive.  A similar failure also occurs in the case of Beer-Aroma, where the model selects texts with a strong sentiment but uncorrelated with the predefined ground truth.

---

> > ### Author Response · Authors · 2022-08-09
> > **Thanks for reviewing the paper**
> >
> > Thank you very much for taking the time to review our paper! With best wishes to you and yours!

---

> ### Author Response · Authors · 2022-08-03
> **Part1-Discussion about specialized layers**
>
> Q1. The paper did not thoroughly discuss the discrepancy when using a unified encoder. The functions of two components are related but not identical. While parts of the encoder can be shared indeed, I cannot see why the unified encoder can work well if there are no other specialized layers in each component's encoder. Have you considered adding specialized layers in each component's encoder such that the encoders are partly shared
> but not identical.
>
> A1. Thank you very much for your suggestion. Actually we have conducted some experiments on the encoders which are partly shared, but did not add it to the paper due to the space limit.
>
> To make a fair comparison with the baseline models, our original paper mainly implemented the encoder with only one layer of GRUs for the experiments, which keeps the same as them.
>
> To better understand the behavior where the encoders are partly shared, we further constructed the encoders with 3 layers of GRUs. The results are shown in the following table.
>
> | Beer-Aroma | s    | acc  | p    | r    | f1    |
> | ---------- | ---- | ---- | ---- | ---- | ---- |
> | all(FR)    | 15.3 | 86.7 | **75.2** | **74.5** | **74.9** |
> | 1          | 14.4 | 83.5 | 68.2 | 62.9 | 65.4 |
> | 1+2        | 15.2 | 84.8 | 75.2 | 73.2 | 74.2 |
> | 3          | 15.5 | 82.4 | 64.2 | 63.9 | 64.1 |
> | 2+3        | 14.6 | 83.9 | 74.1 | 69.5 | 71.7 |
> | 0(RNP)     | 15.7 | 82.9 | 63.4 | 64.0   | 63.7 |
>
> | Hotel-Cleanliness | s    | acc  | p    | r    | f1    |
> | ----------------- | ---- | ---- | ---- | ---- | ---- |
> | all(FR)           | 11.2 | 97.0   | **34.3** | **43.4** | **38.3** |
> | 1                 | 10.1 | 96.0   | 7.5  | 7.7  | 7.6  |
> | 1+2               | 10.4 | 97.0   | 20.2 | 23.7 | 21.8 |
> | 3                 | 9.7  | 97.0   | 32.5 | 35.7 | 34.0   |
> | 2+3               | 10.0   | 97.5 | 32.6 | 36.8 | 34.5 |
> | 0(RNP)            | 9.8  | 97.0   | 9.0    | 9.9  | 9.4  |
>
> Notes:
>
> All (FR): the unified encoder (all 3 layers are fully shared)
>
> 1: only the 1st layer is shared
>
> 1+2: the 1st and 2nd layers are shared
>
> 3: only the 3rd layer is shared
>
> 2+3: the 2nd and 3rd layers are shared
>
> 0(RNP): all 3 layers are not shared
>
>
> It can be seen from the table that the case fully sharing the encoder between the generator and predictor always outperforms that where the encoders are partly shared in terms of rationale quality (f1).
> Besides, the above table showed that the rationale quality gets better as the number of shared layers increases in overall.
>
>
> We argue that, in order to achieve a good rationale, both the generator and predictor have to capture and encode the same informative sections from an input as discussed in lines 152-175 of the paper. Therefore,
> sharing the entire encoder can help performance. We also showed in the theoretical analysis of the paper, the encoder should be fully shared to make the predictor and the generator regularized by each other to get stable models.
>
> Your suggestion on adding analysis on partly shared encoders is really helpful, and it does help the reader to understand better what has inspired our idea of the unified encoder,  and we will add this part to the updated version of our paper.

---

> > ### Author Response · Authors · 2022-08-07
> > **More experimental results on encoders with different numbers of shared layers**
> >
> > We recently conducted more experiments on encoders with different numbers of layers ranging from $2$ to $5$. We adopt the Beer-Aroma dataset and the results are presented in the following tables.
> >
> >
> >
> >
> > |  2-layer | s     | acc   | p     | r     | f1    |
> > |-------------|-------|-------|-------|-------|-------|
> > | none(RNP)        | 16.7  | 78.7  | 50.8  | 54.5  | 52.6  |
> > | 1st         | 15.3  | 82.3  | 68.2  | 67.0  | 67.6  |
> > | 2nd         | 14.8  | 84.0  | 63.7  | 60.6  | 62.1  |
> > | all(FR)         | 14.5  | 86.2  | **77.6** | **72.0**  | **74.7**  |
> >
> > Notes:
> >
> > none(RNP): all two layers are not shared
> >
> > 1st: the first layer is shared
> >
> > 2nd: the second layer is shared
> >
> > all(FR): all two layers are shared
> >
> >
> >
> >
> > | 4-layer | s     | acc   | p     | r     | f1     |
> > |------------|-------|-------|-------|-------|-------|
> > | 0(RNP)          | 14.1  | 81.2  | 73.2  | 66.6  | 69.4  |
> > | 1          | 16.7  | 85.1  | 68.0  | 73.1  | 70.5  |
> > | 2          | 14.8  | 84.7  | 72.8  | 69.1  | 70.9  |
> > | 3          | 14.9  | 82.3  | 75.5  | 72.3  | 73.9  |
> > | all(FR)        | 15.1  | 88.0  | **76.8**  | **74.3**  | **75.1**  |
> >
> > Notes:
> >
> > 0(RNP): all four layers are not shared
> >
> > 1: the first layer is shared
> >
> > 2: the first two layers are shared
> >
> > 3: the first three layers are shared
> >
> > all(FR): all four layers are shared
> >
> >
> >
> > | 5-layer | s     | acc   | p     | r     | f1     |
> > |------------|-------|-------|-------|-------|-------|
> > | 0(RNP)          | 17.1  | 85.8  | 66.9  | 75.7  | 71.0  |
> > | 1          | 14.3  | 83.6  | 74.3  | 68.4  | 71.2  |
> > | 2          | 14.7  | 85.2  | 74.2  | 70.1  | 72.1  |
> > | 3          | 15.6  | 85.6  | 71.0  | 71.0  | 71.0  |
> > | 4          | 16.3  | 87.7  | 71.9  | **75.2**  | 73.5  |
> > | all(FR)        | 15.4  | 87.3  | **75.2**  | 74.2  | **74.7**  |
> >
> >
> > Notes:
> >
> > 0(RNP): all five layers are not shared
> >
> > 1: the first layer is shared
> >
> > 2: the first two layers are shared
> >
> > 3: the first three layers are shared
> >
> > 4: the first four layers are shared
> >
> > all(FR): all five layers are shared
> >
> >
> > As can be seen, these results show similar phenomena to the previous results on the 3-layer encoder, from which the same conclusion can be drawn.

---

> > > ### Author Response · Authors · 2022-08-08
> > > **Results have beed uploaded to Appendix D**
> > >
> > > We have uploaded all these experimental results to the new Appendix D. We sincerely look forward to hearing from you. If you have any questions, please do not hesitate to discuss them with us.

---

> > > > ### Comment · Reviewer_oaM2 · 2022-08-09
> > > > **Response received**
> > > >
> > > > Thank you for the response and added experiments. My questions have been addressed. I will keep my original score.

---

### Author Response · Authors · 2022-08-07
**Summary of main concerns from all the reviewers and the corresponding reponses**

For ease of reading, we here make a summarization of the main concerns of all reviewers and give correspondingly summarized responses.

**Q1**(Reviewer oaM2) The paper did not thoroughly discuss the discrepancy when using a unified encoder. Have you considered adding specialized layers in each component's encoder such that the encoders are partly shared but not identical.

**A1**. To demystify this problem, we now added more experiments on the encoders of which partial layers are shared. The experimental results show that sharing the full encoder always performs the best, which keeps consistent with our theory. Besides, the results showed that the overall performance gets better as the number of shared layers increases. Detailed results and more discussions can be found in the responses to Reviewer oaM2.




**Q2**(Reviewer 7cgR) Definition 1 for uninformativeness doesn't hold. Considering the word "not", it is uninformative by the definition. But "not good" and "not bad" might be meaningful collocations. This case breaks Lemma 1: the behavior of the model is changed by inserting "not".

**A2**. This issue arises because the statement of Definition 1 is not very rigorous, resulting in misleading. To address this issue, we in the revised paper provide a more formal and rigorous statement of Definition 1. In fact, the revised statement has been informally described in lines 188-189 of the original paper, on which Lemma 1 is derived.


**Q3**(Reviewer 7cgR) I don't see how Lemma 3 holds for cases like this with a well-trained encoder. (I don't see derivations of either lemma from first principles; these are simply asserted, and I disagree with them as they are asserted). This state of affairs is worse for pre-trained Transformer models, where the holdover from the pre-training objective makes much of the reasoning about the objective for this particular problem moot anyway, as the encoder's behavior reflects the base language modeling objective as well.
Even setting aside collocations, I am also skeptical of Lemma 3 based on the mathematics of the models themselves. I would believe there are some idealized GRUs that this can be true for, but has this been verified on GRUs for real problems? I think the burden of proof is on the authors to show that there isn't some small latent state update (e.g., counting "uninformative" words) going on at each timestep. Otherwise, if the forget gates are nonzero even for some positions in the latent state vector, this lemma won't hold.

**A3**. The derivation process in our original paper is simplified to some extent due to the space limit, and thus it may lose some readability. For ease of understanding the derivations of Lemma 3 from Lemma 2 and Lemma 1, we further refine the reasoning process in the revised paper and present it in the corresponding response. Besides, for all the concerns about Lemma 3, we identify that there is a major misunderstanding about our theoretical analysis. Our theory is not to analyze the intermediate states of the model, e.g., a not converged model in the training process or a pre-trained model going to be finetuned, but the properties of the model after it has got the optimal solution for the loss function of rationalization (refer to equation 2 in the original paper) as specified in lines 181-182 and lines 196-197. To this end, how to initialize the model does not affect the theoretical properties of the final model. Besides, we also did theoretical analysis for Transformer encoders, as shown in Appendix A.2. Furthermore, we conducted extensive experiments where we pre-train the generator and the predictor with special loss functions to deliberately induce degeneration, and the results can be found in Table 5 and Table 6. We also conducted experiments with Bert of which the results are presented in the response A5 for Reviewer 7cgR. Both results show that our model still outperforms the baselines under the case where the pre-trained models are leveraged.

To verify Lemma 3 for GRUs on practical problems, we give the visualization of some examples for Lemma 3 in the course of the experiments. The results are presented in the new Appendix B.5, which directly supports Lemma 3. A detailed discussion of experimental results can be found in Appendix B.5.



Finally, we sincerely look forward to hearing from the reviewers. If you have any questions, please do not hesitate to discuss them with us.

---

### Meta-Review · Area_Chair_HwHP · 2022-08-28

**Recommendation:** Accept
**Confidence:** Certain

**Metareview:**

There is consensus between reviewers that this is a worthwhile paper suitable for this venue.  I also apppreciate the extensive back and forths between the authors and the reviewers that seem to have improved the paper during the reviewing period.

**Award:**

No

---

### Decision · Program_Chairs · 2022-09-14

Accept